# Understanding the uncertainty in global forest carbon turnover

Thomas A. M. Pugh[1,2], Tim Rademacher[2,4,5], Sarah L. Shafer[6], Jörg Steinkamp[7,8], Jonathan Barichivich[9,10], Brian Beckage[11], Vanessa Haverd[12], Anna Harper[13], Jens Heinke[14], Kazuya Nishina[15], Anja Rammig[16], Hisashi Sato[17], Almut Arneth[18], Stijn Hantson[19], Thomas Hickler[7,20], Markus Kautz[21], Benjamin Quesada[18,22], Benjamin Smith[23,24], Kirsten Thonicke[14]

[1] School of Geography, Earth & Environmental Sciences, University of Birmingham, Birmingham, B15 2TT, United Kingdom.
[2] Birmingham Institute of Forest Research, University of Birmingham, Birmingham, B15 2TT, United Kingdom.
[3] Department of Organismic and Evolutionary Biology, Harvard University, Cambridge, MA 02138, USA.
[4] School of Informatics, Computing and Cyber Systems, Northern Arizona University, Flagstaff, AZ 86011, USA.
[5] Center for Ecosystem Science and Society, Northern Arizona University, Flagstaff, AZ 86011, USA.
[6] Geosciences and Environmental Change Science Center, U.S. Geological Survey, 3200 SW Jefferson Way, Corvallis, OR 97331, USA.
[7] Senckenberg Biodiversity and Climate Research Centre (BiK-F), Senckenberganlage 25, 60325 Frankfurt/Main, Germany.
[8] Johannes Gutenberg-University Mainz, Anselm-Franz-von-Bentzel-Weg 12, 55128 Mainz, Germany.
[9] Instituto de Conservación Biodiversidad y Territorio, Universidad Austral de Chile, Valdivia, Chile.
[10] Laboratoire des Sciences du Climat et de l'Environnement, IPSL, CNRS/CEA/UVSQ, 91191 Gif sur Yvette, France.
[11] Department of Plant Biology & Department of Computer Science, University of Vermont, Burlington, VT 05405, USA.
[12] CSIRO Oceans and Atmosphere, PO Box 3023, Canberra, ACT 2601, Australia.
[13] College of Engineering, Mathematics, and Physical Sciences, University of Exeter, Exeter, UK
[14] Potsdam-Institute for Climate Impact Research (PIK), Telegraphenberg, 14473 Potsdam, Germany.
[15] Center for Regional Environmental Studies, National Institute for Environmental Studies (NIES), 16-2, Onogawa, Tsukuba, Japan
[16] Technical University of Munich (TUM), School of Life Sciences Weihenstephan, Freising, Germany.
[17] Institute of Arctic Climate and Environment Research (IACE), Japan Agency for Marine-Earth Science and Technology (JAMSTEC), 3173-25 Showamachi, Kanazawa-ku, Yokohama, 236-0001, Japan.
[18] Karlsruhe Institute of Technology, Institute of Atmospheric Environmental Research (IMK-IFU), Kreuzeckbahnstrasse 19, 82467, Garmisch-Partenkirchen, Germany.
[19] Department of Earth System Science, University of California, Irvine, CA, USA.
[20] Department of Physical Geography, Goethe University, Altenhöferallee 1, 60348 Frankfurt/Main, Germany.
[21] Department of Forest Health, Forest Research Institute Baden-Württemberg, 79100 Freiburg, Germany.
[22] Universidad del Rosario, Faculty of Natural Sciences and Mathematics, Research Group "Interactions Climate-Ecosystems (ICE)", Cra 26 63b-48, 111221, Bogotá, Colombia
[23] Department of Physical Geography and Ecosystem Science, Lund University, 22362 Lund, Sweden.
[24] Hawkesbury Institute for the Environment, Western Sydney University, Locked Bag 1797, Penrith NSW 2751, Australia.

Correspondence to: Thomas A. M. Pugh (t.a.m.pugh@bham.ac.uk)

**Abstract.** The length of time that carbon remains in forest biomass is one of the largest uncertainties in the global carbon cycle, with both recent-historical baselines and future responses to environmental change poorly constrained by available observations. In the absence of large-scale observations, models used for global assessments tend to fall back on simplified assumptions of the turnover rates of biomass and soil carbon pools. In this study, the biomass carbon turnover times calculated by an ensemble of contemporary terrestrial biosphere models (TBMs) are analysed to assess their current capability to accurately estimate biomass carbon turnover times in forests and how these times are anticipated to change in the future.

Modelled baseline 1985-2014 global average forest biomass turnover times vary from 12.2 to 23.5 years between TBMs. TBM differences in phenological processes, which control allocation to, and turnover rate of, leaves and fine roots, are as important as tree mortality with regard to explaining the variation in total turnover among TBMs. The different governing mechanisms exhibited by each TBM result in a wide range of plausible turnover time projections for the end of the century. Based on these simulations, it is not possible to draw robust conclusions regarding likely future changes in turnover time, and thus biomass change, for different regions. Both spatial and temporal uncertainty in turnover time are strongly linked to model assumptions concerning plant functional type distributions and their controls. Thirteen model-based hypotheses of controls on turnover time are identified, along with recommendations for pragmatic steps to test them using existing and novel observations. Efforts to resolve uncertainty in turnover time, and thus its impacts on the future evolution of biomass carbon stocks across the world's forests, will need to address both mortality and establishment components of forest demography, as well as allocation of carbon to woody versus non-woody biomass growth.

## 1 Introduction

Large uncertainties persist in the magnitude and direction of the response of the terrestrial carbon cycle to changes in climate, atmospheric $CO_2$ concentration, and nutrient availability (Ciais et al., 2013; Friedlingstein et al., 2014), which prevent definitive statements on carbon cycle-climate feedbacks (Arneth et al., 2010; Ciais et al., 2013). Carbon uptake and turnover by forests is a very large component in the global carbon cycle on the scale of decades to centuries (Carvalhais et al., 2014; Jones et al., 2013; Pugh et al., 2019a). The gain or loss of carbon in terrestrial ecosystems is a function of net carbon input to the system, via net primary productivity (NPP), and the rate of carbon turnover (loss) in the system. For vegetation this can be formalised as:

$$dC_{veg}/dt = NPP - F_{turn} = NPP - C_{veg}/\tau \qquad \text{(Eq. 1),}$$

where $C_{veg}$ is the stock of carbon in live biomass and $\tau$ the mean turnover time of that live biomass, i.e. the mean time that carbon remains in living vegetation. Turnover time of existing biomass can thus be calculated as,

$$\tau = C_{veg}/F_{turn} \qquad \text{(Eq. 2),}$$

(Sierra et al., 2017). $F_{turn}$ is the total loss flux of live biomass due to the transfer of plant tissue to dead pools of litter and soil, to harvest products and residues, or to the atmosphere via burning. It can be decomposed into its major components,

$$F_{turn} = F_{mort} + F_{leaf} + F_{fineroot} + F_{repro} \qquad \text{(Eq. 3),}$$

where $F_{mort}$ is the carbon turnover flux due to plant mortality or woody carbon loss, $F_{leaf}$ and $F_{fineroot}$ that due to leaf and fine root senescence respectively, and $F_{repro}$ turnover due to reproductive processes (e.g. flowers, fruits). Neither NPP nor $\tau$ are constant but are affected by many factors including climate, physiological stress, disturbances, species, functional group or ecosystem type. Relatively little attention has focused on the representation of $\tau$ and its drivers in current vegetation models, with some but not all relevant dependencies represented in different models. Until recently, most attention has instead focussed on understanding spatial and temporal dynamics of NPP and respiration carbon losses (e.g. Ahlström et al., 2015a, 2012; Ballantyne et al., 2017; Cramer et al., 1999; Schaphoff et al., 2006). Recently, however, a number of studies have found $\tau$ to have comparable or even larger importance than NPP when assessing the response of $C_{veg}$ to environmental change using terrestrial biosphere models (TBMs) (Ahlström et al., 2015a; Friend et al., 2014; Galbraith et al., 2013; Johnson et al., 2016; Thurner et al., 2017), with large divergence in TBM projections of $\tau$ over the 21$^{st}$ century depending on forcing (Ahlström et al. 2015a) or the choice of TBM (Friend et al. 2014). The divergence that can be traced to TBM structure and parameterisation (Nishina et al., 2015) has not been closely analysed in terms of the contributions of specific underlying processes, interactions and driver dependencies, or their basis in knowledge from real world ecosystems.

Conceptually, turnover time of carbon in live vegetation is a function of carbon allocation to biomass pools with different characteristic turnover times, and changes in these turnover times in response to environmental variation. TBMs typically aim to represent the landscape across hundreds or thousands of square kilometres. At this scale, not only individual plant behaviour, but also changes in the functional species composition, affect $\tau$. Under environmental change, there are several mechanisms by which $\tau$ and biomass may be altered (Table 1). Thus, effects of environmental change on $\tau$ can be divided into three groupings, those associated with changes to allocation patterns of individual trees within the current mix of species (denoted MI in Table 1), those associated with collective responses of multiple individuals at the stand level (MS) and those associated with a population-level change in species mix (MP). Mechanisms within these groupings are distinguished in Table 1 so as to show how a particular perturbation in NPP, allocation, or turnover rate of woody or soft tissues (e.g. leaves, fine roots and fruits) would affect biomass or $\tau$. Because trees and ecosystems respond to environmental stimuli in a coordinated fashion, it is likely that many of these mechanisms will occur in concert.

Most carbon in forest vegetation is stored in wood which has relatively long turnover times compared to soft tissues. Turnover of wood is believed to primarily result from tree mortality, although branchfall also occurs yet is poorly quantified (Marvin and Asner, 2016). Natural mortality in trees can have many causes, including both primarily biotic (e.g. competition, insects, senescence) and abiotic (e.g. fire, drought, windthrow) causes, and often involves complex interactions with forest structure (Brando et al., 2014; Franklin et al., 1987). Compared to productivity, quantitative understanding of tree mortality is at a fledgling stage, with large unknowns relating to different processes of death and their environmental dependencies (Anderegg et al., 2016; Hartmann et al., 2018; McDowell et al., 2008; Sevanto et al., 2014). Accordingly, neither plant-physiological

processes nor interactions of multiple stresses are represented in great detail in current TBMs, although some aspects of the hydraulic and carbohydrate system, and coupled carbon- and water-related physiology, may be linked to mortality in these models. As reviewed in McDowell et al. (2011) and Adams et al. (2013) (see also Section 2.2 herein), TBMs often prescribe

bioclimatic limits for establishment and survival, or threshold temperatures combined with how often the threshold is exceeded to determine mortality. Mortality is triggered in some models by a negative carbon balance or if tree vigour is low (for instance, if growth efficiency, the ratio of NPP to leaf area, falls below a defined threshold; Smith et al., 2001). In principle, such formulations should capture both environmental stress and competition with neighbours, but in some TBMs such processes are supplemented or replaced by self-thinning rules to represent this typical effect of size-dependent competition in densifying

stands (e.g. Haverd et al., 2014; Sitch et al., 2003). Here we refer to all such mechanisms related to carbon balance, vigour or competition as "vitality-based". Mortality in association with disturbance, such as storms or insect outbreaks, are captured in some TBMs by a set "background" mortality, the likelihood of which may be size or age related (e.g. Smith et al., 2014). Wildfires are now included as a dynamic process in many TBMs; however the representation of the impact of fire on vegetation structure is still immature (Hantson et al., 2016). Ultimately, the effect of a change in mortality rate on $\tau$ may be either direct

(Table 1, $MI_{MR}$), or indirect, via shifts in tree functional composition (possibly mediated by $MI_{MR}$) that change the mean behaviour of the tree population at the landscape scale (MP).

As for wood, turnover rates of soft tissues due to phenological cycles also lack strong constraints, with fine root turnover being challenging to measure (Lukac, 2012) and reproductive investment differing widely with species and life stage (Wenk and

Falster, 2015). Leaf cover dynamics are readily observed, e.g. from satellite data, but turnover rates can be difficult to ascertain, particularly in evergreen trees, and can vary due to plant-external factors such as herbivory. Although the carbon stock in soft tissues may be relatively small compared to wood, these phenological turnover rates influence the amount of carbon that trees must allocate to maintain a given leaf area or root network, affecting how much carbon is left over to produce wood. In this way, uncertainties in phenological turnover rates will influence overall biomass $\tau$ in TBMs. Allocation patterns within a given

plant or plant type may also change as a function of environmental conditions ($MI_{RA}$), for instance based on a "functional balance" principle in which resources are allocated to alleviate the most limiting constraint(s) at a given point in time (Franklin et al., 2012; Sitch et al., 2003). Models in which vegetation composition is able to evolve with climate often include effective allocation shifts at the population level in calculations of $\tau$ ($MP_{RA}$). Overall, changes in phenological turnover rates, either at the individual level ($MI_{ST}$), or through vegetation composition shifts ($MP_{ST}$) may have profound influences on $\tau$.


Changes in productivity affect biomass accumulation ($MI_{NPP,F}$, $MP_{NPP}$) but do not affect $\tau$ directly. However, they may accelerate the self-thinning process ($MS_{comp}$) and also change mortality rate through the link to tree vitality. Furthermore, if changes in productivity are accompanied by an allocation response, for instance a reduced allocation to leaves and stems in favour of roots as soil resources become limiting ($MI_{NPP,FS}$), then $\tau$ will be impacted.

Here, an ensemble of six representative current TBMs (Table 2) was analysed to compare the mechanisms they encapsulate governing vegetation carbon turnover and its impacts on modelled carbon pools and fluxes (Table 3). Expanding on previous work (e.g. Friend et al., 2014), the aims were to:

1) assess the baseline variation in $\tau$ within and between TBMs and identify the reasons for these variations;

2) evaluate the simulated $\tau$ and its components against existing observations where available;

3) diagnose why projections of future $\tau$ diverge between models;

4) identify model-based hypotheses for the spatial and temporal variation in $\tau$ to guide future research to quantify and predict terrestrial carbon cycling.

We first analyse historical vegetation carbon turnover time estimates from the models, comparing the models with available

large-scale observations and identifying implicit or explicit model-based hypotheses that may explain why the estimates diverge (Section 3.1). We then identify hypotheses behind differing future turnover time estimates under an exemplary climate change scenario (Section 3.2). Finally, we discuss how these hypotheses can be tested to advance understanding of turnover times, building on available data sources where possible (Section 4). Our analysis is restricted to forests, which contain the vast majority of vegetation carbon (Carvalhais et al., 2014). Land-use change and management has profoundly changed

biomass turnover rates over the last centuries (Erb et al., 2016), but is disregarded here in order to focus attention on the intrinsic dynamics of forests. Dynamic changes in vegetation composition driven by dispersal and migration are included, but only within the area currently defined as forest.

## 2. Methods

### 2.1 Definition of $\tau$

The concept of $\tau$ adopted in this study is that presented in Eq. 2, henceforth referred to $\tau_{turn}$. However, $\tau$ is often approximated by $C_{veg}/NPP$ (henceforth $\tau_{NPP}$) (Erb et al., 2016; Thurner et al., 2017), based on the assumption that the system is in pseudo-equilibrium, and therefore $F_{turn} = NPP$ in the multiannual mean. Even in a system under transient forcing, at the global level $\tau_{NPP}$ is likely a close approximation of $\tau_{turn}$ (see results in Table 4). Generally, our analysis focuses on $\tau_{turn}$ because it directly represents turnover, apart from in Fig. 1, where $\tau_{NPP}$ is shown for consistency with the satellite-based data to which the model

estimates are being compared. Where the difference between $\tau_{NPP}$ and $\tau_{turn}$ is of minimal consequence, $\tau$ is used for simplicity. Turnover times can also be defined relative to particular turnover fluxes, such as those outlined in Eq. 3. In this case the turnover time is calculated respective to the appropriate biomass pool, i.e. turnover time of vegetation biomass due to mortality, $\tau_{mort,}$ is defined as $C_{veg}/F_{mort}$, and turnover time of fine root biomass, $\tau_{fineroot,}$ is defined as $C_{fineroot}/F_{fineroot}$, where $C_{fineroot}$ is the fine root biomass. $F_{mort}$ can also be decomposed further into fluxes resulting from particular mortality processes, for instance,

following the conceptual groupings in Table 3,

$$F_{mort} = F_{mort,vitality} + F_{mort,disturbance} + F_{mort,background} + F_{mort,heat} + F_{mort,other} \quad \text{(Eq. 4)}$$

Accordingly, a turnover time can also be defined for $C_{veg}$ relative to each mortality process, e.g. $\tau_{mort,vitality} = C_{veg}/F_{mort,vitality}$. Turnover rates are the inverse of turnover time, i.e. $1/\tau$.

## 2.2 Model descriptions

The TBMs in this study (Table 2) have been widely applied in studies of the regional and global terrestrial biosphere and used in major international assessments (Jones et al., 2013; Le Quéré et al., 2018; Sitch et al., 2008). They simulate the fluxes of carbon between the land surface and the atmosphere, and the cycling of carbon through vegetation and soils. All models simulate the stocks of, and fluxes to and from, wood, leaves and fine roots. A representative range of alternate modelling approaches are encapsulated in this ensemble. Three of the models adopt area-based, average-individual approaches to vegetation representations (LPJmL3.5, ORCHIDEE, JULES), two a cohort-based approach (LPJ-GUESS, CABLE-POP), and one an individual-based approach (SEIB-DGVM). LPJ-GUESS includes a coupled carbon-nitrogen cycle, while all except CABLE-POP include dynamic changes in plant functional type (PFT) composition in response to environmental conditions. The number and type of PFTs vary between the models and are summarised in Table S1. As a group, these models encapsulate many of the mortality process representations currently found in different TBMs (Table 3). Parameters relating to phenological turnover rate are summarised in Table S2.

## 2.3 Model experiments

Two simulations were completed by each TBM: a historical 1901-2014 simulation, driven by the CRU-NCEP v5 observation-based climate product and observed atmospheric $CO_2$ mixing ratios (Le Quéré et al., 2015); and a historical-to-future 1901-2099 simulation driven by climate output fields from the IPSL-CM5A-LR climate model under an RCP 8.5 future scenario, bias-corrected against the observation-based WATCH dataset, as described in Hempel et al. (2013). Deposition of reactive nitrogen species (LPJ-GUESS only) was forced by data from Lamarque et al. (2013). Simulations were of potential natural vegetation (i.e. no anthropogenic land-use was applied), with the exception of CABLE-POP which uses prescribed vegetation cover fractions and thus landcover for the year 1700 was applied. CABLE-POP also differed from the other models in using the CRU-NCEP v7 data set for the historical climate run. Model-standard methods for spin-up were applied, with spin-up $CO_2$ mixing ratio and nitrogen deposition fixed at 1901 values. All simulations were performed at 0.5° × 0.5° grid resolution, with the exception of JULES, which used an 1.875° × 1.25° grid cell size.

In addition to commonly used variables such as NPP, leaf area index (LAI) and $C_{veg}$ for wood, leaves and fine roots, all TBMs also outputted separately the fluxes of carbon turnover from leaf and fine root turnover, and from each individual mortality

process within the model (with the exception of ORCHIDEE, which provided all mortality-driven turnover as a single value). For display purposes, these processes were grouped as described in Table 3. For those models that include a loss of carbon due to reproduction, this was either output directly, or calculated in postprocessing as 10% of NPP, consistent with the given model's assumptions. Unless otherwise stated, results are presented as statistics over a 30-year period, which is 1985-2014 in the baseline case.

## 2.4 Analysis

**Forest masking**: A mask defining forest was developed for each TBM and used for subsequent analyses. For maps of TBM output, values were displayed if (1) the TBM simulated forest for a grid cell and (2) observations for the year 2000 showed the grid cell to contain at least 10% cover of closed-canopy forested area. For calculating regional sums and statistics of TBM output, the second step was implemented by multiplying the TBM output for a grid cell by the observed closed-canopy forested area in that grid cell before calculating statistics. This process results in sums and statistics for each model being calculated over a slightly different area but avoids turnover statistics for forest being skewed e.g. where a TBM erroneously simulates grassland where satellite observations indicate forest. Forest distribution maps for simulations and observations and their discrepancies are shown in Fig. S1.

The masks identifying grid cells where each TBM simulated forest were based on simulated PFT maximum annual LAI values modified by PFT cover fraction for each grid cell. Following Hickler et al. (2006) and Smith et al. (2014), a grid cell was defined as 'forest' in a given year if (a) the maximum annual LAI value summed for all simulated tree PFTs was > 2.5 or (b) the maximum annual LAI value summed for all simulated tree PFTs was > 0.5 and the PFT with the maximum LAI for the grid cell was a boreal tree PFT (i.e., boreal needleleaved evergreen, boreal needleleaved deciduous, or boreal broadleaved deciduous). For JULES and CABLE-POP, which did not break out PFTs into boreal and temperate categories, needleleaved evergreen, needleleaved deciduous, and broadleaved deciduous tree PFTs were considered potential boreal PFTs for step (b). Either condition (a) or (b) needed to be satisfied for at least 10 years during the period 1985-2014 for the grid cell to be assigned as forest.

To only consider recent-historical forest areas, forest masks were further constrained based on year 2000 satellite remote-sensing of forest cover following Pugh et al. (2019a). Forest cover at ca. 30 × 30 m (Hansen et al., 2013) was aggregated to 30 × 30 arc seconds, and designated as closed-canopy forest if canopy coverage exceeded 50% of the aggregated grid cell. Percentage closed-canopy forest coverage was then calculated for each 0.5° × 0.5° grid cell (each 1.875° × 1.25° grid cell for JULES). Grid cells with less than 10% closed-canopy forest cover by this definition are not displayed on the maps, but data from these grid cells are included in the global and regional sums and statistics for the TBMs.

**Observation-based forest type classification:** Forest type was defined as in Pugh et al. (2019b) based on the latest landcover product from the European Space Agency (ESA, 2017). The mapping of ESA landcover classes to the forest types is summarised in Table S3 and the resulting forest-type distribution is shown in Fig. S2.

**Model forest type classification:** To facilitate analysis of changes in forest composition, PFTs were classified into seven forest types (Table S1) based on phenological traits. LAI (1985-2014, 30-year mean) for all the PFTs within each forest type was summed, and the grid cell was assigned a forest type according to the grouping with the highest LAI sum. This process produced a forest-type mask for each model (Fig. S1). The unification of forest types across models means that each forest type may be composed of 1-3 PFTs.

**Satellite-based estimates of $\tau_{NPP}$:** Satellite-derived biomass and NPP products allow $\tau_{NPP}$ to be estimated as described in Section 2.1. Here, estimates were made for all grid cells with at least 10% closed-canopy forest cover. A contemporary product of total (above- and below-ground) vegetation carbon as prepared by Carvalhais et al. (2014), based on Saatchi et al. (2011) and Thurner et al. (2014), was used. In order to be comparable with the TBM simulations, this observational biomass product was corrected for landcover by dividing the biomass values by closed-canopy forest area, making the assumption that biomass outside closed-canopy forests is negligible. NPP for the same period was estimated by averaging the MODIS NPP (Zhao and Running, 2010) and BETHY/DLR (Tum et al., 2016; Wißkirchen et al., 2013) products over the period 2000 to 2012 as per Thurner et al. (2017), making the assumption that NPP was uniform across each grid cell.

**Tropical $\tau_{mort}$ evaluation:** For South America, plot-level observations of above ground biomass (AGB) and turnover rate of AGB due to mortality were taken from Brienen et al. (2014, 2015). Mean values of AGB and AGB turnover rate were calculated across all census intervals at each of 274 plots. These data were summarised into a plot-mean $\tau_{mort}$, weighting each census equally and assuming that $\tau_{mort}$ of AGB and total biomass are equivalent. For Africa and Asia/Australia, plot data were taken from Galbraith et al. (2013). For each plot, the modelled value of $\tau_{mort}$ was extracted for the grid cell in which the plot was located, creating a vector of modelled $\tau_{mort}$ with the same spatial weighting as in the observations. Modelled $\tau_{mort}$ for each plot was a mean over the years between the beginning of the first census and end of the last census at that plot for the South American data, and over 1985-2014 for the other data, for which census interval information was not provided. Equivalent compilations for temperate and boreal zones were not available.

**Drought-mortality evaluation:** Very limited information on large-scale tree mortality due to extreme events is currently available for evaluating model simulations. Here, the TBMs forced by CRU-NCEP were compared to drought-related tree mortality observed at a number of sites (Allen et al., 2010, as summarised by Steinkamp and Hickler, 2015). The fraction of sites for which each TBM simulated a significant increase in mortality in the 5 years following the observed drought-mortality

event, relative to the whole simulation, was calculated with a Wilcoxon Rank Test on mortality fluxes using a 5% significance level. This fraction was compared against a likelihood of 10 randomly selected 5-year intervals seeing significantly enhanced mortality. For each TBM, only observed data from sites where the TBM simulated forest (as defined by the forest mask for each TBM) were considered.

**Contribution of turnover fluxes to spatial variation in $\tau$:** Following Eqs. 2 and 3, $\tau_{turn} = C_{veg}/(F_{mort} + F_{phen})$, where, $F_{phen} = F_{leaf} + F_{fineroot} + F_{repro}$. $\tau_{turn}$ was calculated for each grid cell with at least 10% forest cover. $\tau_{turn,fixmort}$ was then calculated in the same way except for replacing the local value of $F_{mort}$ with its mean across all grid cells. The difference between $\tau_{turn}$ and $\tau_{turn,fixmort}$ provides the difference in $\tau_{turn}$ due to the local deviation in $F_{mort}$. The results were summarised at the global level by taking the mean absolute deviation of $\tau_{turn} - \tau_{turn,fixmort}$ across all grid cells. The same procedure was carried out to assess deviation due to $F_{phen}$.

## 3. Results

### 3.1 Recent-historical $C_{veg}$ and $\tau$

Simulated total $C_{veg}$ in global closed-canopy forests ranges from 284 to 432 Pg C among models, with two distinct clusters around the extremes of this range (Table 4). Satellite-based $C_{veg}$ over the same area is consistent with the upper end of the range at 450 Pg C, although the satellite-based estimate includes management effects not explicitly included in the model simulations here. There is large variation in the global total of forest NPP between models (Table 4), but consistency in the relative global pattern (Fig. S3). Modelled global mean $\tau_{NPP}$ for forest vegetation varies from 11.9 to 22.6 years, which may be compared to the satellite-based estimate of 19.3 years, although the latter implicitly includes the effects of management. Regional variations can be even more pronounced, for instance $\tau_{NPP}$ varies from *ca.* 10 to 25 years for parts of the Amazon region, and *ca.* 5 to 30 years for parts of the boreal forest, depending on the model (Fig. 1). Particularly marked is a lack of agreement in the relative differences between regions, with four models (CABLE-POP, JULES, LPJ-GUESS, LPJmL) simulating $\tau_{NPP}$ to be longer in tropical forests than in extratropical forests, whereas ORCHIDEE and SEIB-DGVM show a much more mixed pattern (Fig. 1). The satellite-based estimate also finds $\tau_{NPP}$ to be longer in the tropics than the extratropics. Notably, the global frequency distribution of $\tau_{NPP}$ from the satellite-based estimate is unimodal with a strong left-skew and a wide range of $\tau_{NPP}$ found across all forest types (Fig. 2). In contrast, $\tau_{NPP}$ distributions modelled by the TBMs are often multimodal, and in many cases characterised by distributions for individual forest types that only span a fraction of the global range in $\tau_{NPP}$. Relative abundance of forest types also varies substantially between models (Fig. 2, Fig. S7).

Overall, mortality is responsible for 37 to 81% of $F_{turn}$, but is less than 50% of $F_{turn}$ for four of the six models (Fig. 3). Much of this variation comes from fine roots, for which the fraction of $F_{turn}$ varies from 6 to 37% depending on the model, whilst the

fraction of $F_{turn}$ due to leaf phenology varies from 13 to 26% (Fig. 3). Consistent with the logic that $F_{turn} \approx$ NPP (Section 2.1), the partitioning of $F_{turn}$ among tissue types is approximately equal to the allocation of NPP between those tissue types. For no change in overall structure, a fraction of $F_{turn}$ resulting from leaf, fine root or reproductive turnover implies the same fraction of NPP must be invested in the corresponding tissues. Therefore, to maintain a given biomass for a given NPP, the results in Fig. 3 reflect two distinct hypotheses linking allocation of NPP to $\tau_{mort}$. Either a large fraction of NPP is invested into wood, resulting in $F_{mort}$ being a large fraction of $F_{turn}$ and thus implying a relatively low $\tau_{mort}$, or a relatively low fraction of NPP is invested into wood, resulting in $F_{mort}$ being a relatively small fraction of $F_{turn}$ and thus requiring a higher $\tau_{mort}$ in order to maintain the same biomass (Table 5: H1a and H1b).

Consistent with the large fraction of turnover flux resulting from soft tissues, both phenological and mortality turnover fluxes contribute substantially to spatial variation in the turnover flux in all TBMs except JULES (Fig. 4) (Table 5: H2). The substantially different shapes of the probability density distribution for each TBM for $\tau_{mort}$ compared to $\tau_{NPP}$ (Fig. 2 vs. Fig. 5) further illustrate the extent to which phenological processes influence $F_{turn}$ among models.

There are large disparities between the TBMs in terms of the turnover rates assigned to fine roots. For instance, JULES assumes fine root longevities 2-3 times longer than the other models (Table S2), resulting in a global mean fine root carbon turnover time ($\tau_{fineroot}$) of 5.0 years (Table 4), consistent with the very small fraction of $F_{turn}$ realised via fine roots. In contrast, $\tau_{fineroot}$ for CABLE-POP is just 0.6 years. Leaf carbon turnover times for evergreen PFTs also differ notably between TBMs (Table S2). Although the models typically reflect the empirical trade-off of leaf longevity with specific leaf area (Reich et al., 1997), the relationship is not proportional, with substantially more carbon required to maintain a canopy with leaves of one-year longevity compared to two years (Fig. S4). Large differences between the models in leaf cost for a given longevity are also apparent. Finally, the models differ in the amount of biomass required in each tissue type, for instance in the assumed ratio of leaf area to sapwood cross-sectional area (LA:SA). For the models considered here with clearly defined LA:SA (Table S4), the choice of LA:SA influences the maximum LAI simulated. For instance, LPJ-GUESS almost uniformly simulates lower LAI than LPJmL (Fig. S5), in line with the lower LA:SA used. Consistent with these differences in PFT-level parameters, spatial variation in the fraction of turnover due to phenology closely follows forest-type distribution (cf. Fig. S6 and Fig. S7) and spatial variability in phenological turnover flux was higher across than within forest types for five of the models (Fig. S8).

Whilst the phenological turnover flux is crucial for allocation of NPP, much larger carbon stocks are held in wood than in soft tissues. Across five of the models here, the fraction of turnover due to mortality is higher in the tropics than at higher latitudes (Fig. S6; LPJmL shows the opposite behaviour), indicating a greater relative allocation to wood compared to soft tissues in this region. However, mean turnover times due to mortality ($\tau_{mort}$) are much less consistent between models. The tropical broadleaved evergreen forest type is simulated to have the highest mean $\tau_{mort}$ by LPJmL, whilst CABLE-POP and LPJ-GUESS

simulate highest mean $\tau_{mort}$ for needleleaved evergreen forest, JULES for boreal broadleaved deciduous forest and ORCHIDEE for temperate broadleaved evergreen forest (Fig. 5). Greater allocation to wood, higher $\tau_{mort}$, or a combination of both could help account for high tropical forest biomass, and the models reflect these alternative hypotheses (Table 5: H3). Comparison of modelled $\tau_{mort}$ with observations from tropical forest plots suggests that most of the TBMs here may substantially underestimate $\tau_{mort}$ in this region (Fig. 6), suggesting that allocation of carbon to wood in the tropics might be overestimated. As for phenological turnover, spatial variation in mortality turnover flux is closely linked to forest-type distribution (Fig. S8), reflecting PFT-specific mortality thresholds or likelihood functions, or even PFT-specific mortality processes (e.g. heat stress in LPJmL).

The wide spread in $\tau_{mort}$ across models (Table 4) and forest types (Fig. 5) reflects the range of approaches used to represent mortality. Despite this diversity, there are similarities in the broad categories of processes included. All models include a mortality process based on low vitality and five of the models include some kind of mortality from physical disturbance (for instance, fire or a generic random disturbance intended to represent, e.g., wind-throw and biotic disturbance; Table 3). Classifying the models according to the relative importance of conceptually distinct mortality processes reveals markedly different hypotheses as to whether vitality or a physical disturbance is the primary cause of carbon turnover from mortality across global forests (Fig. 7) (Table 5: H4). Latitudinal variation in the dominant mortality process is limited (Fig. 7).

The mortality processes included in the TBMs have a limited ability to capture observed tree mortality attributed to drought. For drought-induced mortality, three of the six models (CABLE-POP, JULES, LPJmL) exhibit a substantially greater occurrence of mortality events at times and locations where such events have been reported in the literature, compared to a set of 10 randomly chosen times at each location (Table S5). All models showed some success in capturing dieback events using representations of processes that are conceptually consistent with drought-induced mortality (Table S5). However, the total percentage of observed events captured is very low, not exceeding 27%.

### 3.2 Future changes in $\tau$ under climate change

The TBMs considered in this study show substantial increases in biomass but divergent responses in $\tau$ over 2000-2099 under projected climate change (Fig. 8), which agrees with the ensemble of Friend et al. (2014). Both negative and positive changes in $\tau_{mort}$ are seen among the simulations (Fig. 8c), but only ORCHIDEE projects an overall global increase in $\tau_{mort}$ over the scenario period. LPJ-GUESS also stands out, displaying a strong decrease in $\tau_{mort}$, despite the strong increase in overall $\tau$. These changes in turnover time show high variability among regions and forest types (Fig. 8), and in several cases clearly follow forest type shifts (Fig. S10). However, the particular mechanisms driving the changes in turnover differ greatly between the models and encompass most of those outlined in Table 1.

Substantial changes in mortality rates (MI$_{MR}$) over 2000-2085 are apparent for at least some forest types in five models (Figs. 9, 10, S9, S11-S16). For example, in temperate broadleaved and needleleaved forests three of the models show increases in vitality-related mortality (JULES, LPJ-GUESS, LPJmL) and one model shows a decrease followed by an increase (CABLE-POP). As described below, the reasons behind these changes differ among models.

In LPJmL, heat stress results in a substantial die-off at the boreal forest southern margin (Table 5: H5a), triggering large, lagged increases in mortality rate due to self-thinning (also a vitality-based mechanism; Table 3) as the young forest regrows (Fig. 9d, S14e-h). The heat-stress mortality rate declines with time as the PFT composition shifts towards temperate broadleaved deciduous trees, which in LPJmL are not subject to heat stress mortality. The substantial changes in mortality rates are thus characteristic of a large-scale dieback and recovery, but are unlikely to be representative of the long-term rates locally once the forest has recovered (see also Sitch et al., 2008). Mortality rates following full recovery from the transition are likely to differ from the pre-transition rates because mortality rates for some processes in LPJmL are PFT specific (MP$_{MR}$), but heat stress mortality remains elevated throughout the 21$^{st}$ century (Fig. S14e-h).

Increases in vitality-induced mortality in LPJ-GUESS (Fig. 9c, S13e-h) show how demographic shifts can result in a change in the mortality rate of a PFT, without any increased likelihood of individual tree death. As the climate warms, the needleleaved PFTs begin to experience establishment failure, and the consequent shift of the age distribution towards larger tree sizes is manifested as an increase in the rate of background mortality of that PFT (likelihood of background mortality is a function of tree age in LPJ-GUESS). As larger trees die, the resulting space is colonised by the shade-intolerant broadleaved deciduous PFT, which is more vulnerable to vitality-induced mortality. Hence, much of the increase in vitality-based mortality is the outcome of, rather than the trigger for, a PFT shift towards a different forest type and an earlier successional stage (MP$_{MR}$). Thus, in LPJ-GUESS, PFT shifts lead to substantial changes in τ through MP mechanisms (Table 5: H5b), but without the same kind of dramatic dieback simulated by LPJmL.

In JULES, increases in vitality-based mortality (Fig. 9b, S12e-h) are the result of ongoing PFT shifts under changing environmental conditions. The growth and loss of carbon due to competition is represented in one equation within JULES, with the most productive PFT being favoured. Changes in mortality rates are thus associated with shifts in forest type, but there are no processes to realise a long-term shift in mortality rates following MI-type mechanisms. Long-term mortality rate shifts can only be realised through MP-type mechanisms (Clark et al., 2011). Thus, JULES implicitly includes a version of hypothesis H5b (Table 5) in that the mortality rate under equilibrium with environmental conditions is independent of those conditions, except to the extent it changes functional composition.

CABLE-POP was run without dynamic vegetation, providing a clear demonstration of processes underlying the $MI_{MR}$ mechanism. The model displays a transient reduction in temperate and needleleaved forest mortality rate in the first half of the 21$^{st}$ century (Fig. 9a, S11e-h) due to increasing NPP, which reduces vitality-induced mortality (Table 5: H6b). The increase in mortality rate towards the end of the 21$^{st}$ century appears to reflect strong warming reducing growth efficiency, possibly related to a temperature-induced reduction in carbon-use efficiency. The self-thinning component of vitality-based mortality increases

throughout the simulation (not shown), as enhanced NPP leads to greater increments in crown size each year, following mechanism $MS_{comp}$ (Table 5: H6a).

In contrast to mortality rate changes in temperate forests, none of the models show large increases in mortality rates across tropical forests, and both LPJmL and ORCHIDEE show substantial decreases in mortality rates in these regions (Fig. S9). For

LPJmL (for which the process breakdown is available; Fig. 10d, S14a-c), this mortality rate decrease appears to be a result of increased NPP reducing the likelihood of growth-efficiency mortality being triggered (Table 5: H6b). However, as all of the models have similar formulations of vitality-based mortality (with the exception of JULES), it is notable that JULES, LPJ-GUESS and SEIB-DGVM show small increases in vitality-induced mortality rates (Figs. 10, S12, S13, S16), alongside strong increases in NPP (Fig. S17). We interpret these results to be further examples of increased mortality through accelerated

resource competition between trees (i.e. self-thinning; $MS_{comp}$, H6a); i.e., although the likelihood of death of the largest trees by vitality-based processes due to environmental extremes may be reduced, turnover rates at the stand level may be maintained or increase as faster growth accelerates competition.

Although the mortality ($MI_{MR}$) and forest-type-shift (MP) mechanisms are important drivers of changes in $\tau$ in the TBMs,

other mechanisms are also relevant in explaining the simulated responses of $\tau$ to environmental change. For instance, LPJ-GUESS displays behaviour following $MI_{NPP,FS}$ (Fig. 8d); as NPP increases, a larger fraction of it is invested in wood (Fig. 3b), increasing $\tau$ despite decreases in $\tau_{mort}$ (Fig. 8b,c). Mechanism $MI_{NPP,FS}$ occurs in all models except ORCHIDEE to varying degrees (Fig. 3b, 8d) (Table 5: H7a), but CABLE-POP and ORCHIDEE tend more towards $MI_{NPP,F}$, which increases biomass with no influence on $\tau$ (Table 5: H7b). LPJ-GUESS and LPJmL reduce their fraction of turnover due to roots more than the

fraction of turnover due to leaves (Fig. 3b). This appears to be a response of the functional-balance allocation approach (Sitch et al., 2003; Smith et al., 2014) to increased water-use efficiency under elevated atmospheric $CO_2$ concentrations ($MI_{RA}$). In contrast, despite encoding a functional-balance approach in which allocation is sensitive to moisture (Krinner et al., 2005), the allocation scheme in ORCHIDEE results in a small increase in the fraction of carbon turnover through roots, perhaps driven by forest-type shifts, and therefore corresponding to $MP_{RA}$.

## 4. Discussion and recommendations

A wide range of estimates of recent-historical and projected future carbon turnover time emerge from the TBM ensemble. As postulated in Table 1, two contrasting modes of simulated turnover response to changing environmental conditions were identified in the simulations: (1) individual or stand-level responses where internal physiology or interactions with neighbours influences turnover in response to temperature, atmospheric $CO_2$ concentration, or other extrinsic drivers (MI, MS mechanisms); and (2) population responses where shifts in species composition influenced forest demography, with concomitant changes to turnover (MP mechanisms). The relative importance of individual, stand and population responses varied across TBMs, as did the processes producing these responses. Of the possible mechanisms governing changes in future $\tau$ and biomass stocks outlined in Table 1, only $MI_{ST}$ and $MP_{NPP}$ could not be clearly identified in the TBM ensemble here. The diversity in both the processes that are included in models (Table 3) and the simulated emergent responses in turnover time, arise largely because the key ecosystem states and fluxes, and their relationships to environmental drivers, are under-constrained by observations at regional and global scales.

Based on the TBM ensemble, several emergent hypotheses (H1-H7) relating to both recent-historical and future carbon turnover rates were identified (Table 5). Resolving the uncertainty around these large-scale carbon turnover rates will require additional observational data, model development, and further testing of the individual hypotheses for different biomes, stand types and environments. In the following discussion, the state of science relating to each hypothesis is briefly reviewed and possible pathways for testing the hypothesis, advancing understanding of turnover times, and reducing TBM uncertainty are suggested.

### 4.1 The partitioning of turnover flux between soft and woody tissues (H1)

Even given firm constraints on biomass and NPP, both forms of hypothesis H1 (H1a and H1b, Table 5) would be possible, necessitating direct constraints on either allocation or turnover rates for soft tissues. Plant trait databases provide numerous observations of leaf longevity and specific leaf area (Kattge et al., 2011). Conversion of this information to typical values at the PFT level should now be possible using species abundance information (e.g. Bruelheide et al., 2018) to appropriately weight species-level data. However, plasticity in plant behaviour, such as leaf shedding during drought or adjustments in specific leaf area under elevated atmospheric $CO_2$ concentrations (Medlyn et al., 2015), requires further investigation, as does the influence of herbivory on leaf turnover, which is usually not considered in TBMs. Using observations to constrain reproductive turnover is more challenging to address; observed investment in reproduction varies between species by up to several tens of percent of NPP, and changes over a tree's life-cycle (Wenk and Falster, 2015). Yet the huge amount of information on seed production (Díaz et al., 2016) is not matched by similar information on fruit and flower production and flowering frequency. Systematic sampling and data compilation efforts to populate knowledge gaps (Wenk and Falster, 2015)

will likely be needed to confidently move beyond assumptions such as the fixed 10% allocation of NPP to reproduction by all vegetation in the LPJ model family (Sitch et al., 2003).

The most striking disparity between models is in the fraction of carbon turned over by fine roots (Fig. 3a). Although some studies have reported turnover times of many years (Matamala and Gonza, 2003), turnover times of around one year or less are supported by meta-analyses for boreal, temperate and tropical forests (Brunner et al., 2013; Finér et al., 2011; Yuan and Chen, 2010), but high methodological uncertainties persist due to inconsistent definitions of fine roots and difficulties in measuring changes in below-ground tissues (Brunner et al., 2013; Finér et al., 2011). In addition, as for leaves, scaling observations across large areas needs to take account of relative species abundances, assuming turnover rates are related to species. Assuming a turnover time of circa one year, fine root production has been estimated to total a third of NPP (Jackson et al., 1997), a larger value than simulated by most of the TBMs included in this study.

Exudates may also use up a substantial percentage of NPP in some ecosystems (Grayston et al., 1996). Conceptually, in TBMs, they may currently be considered as implicit within either fine root allocation or root respiration. Given short turnover times, either assumption is probably adequate as a first approximation, especially when combined with allocation schemes that can capture environmentally driven changes (e.g. functional balance). On-going research, for instance at the current generation of forest free-air $CO_2$ experiments (FACE; Phillips et al., 2011), should provide improved understanding of response functions, allowing better constraints of such responses (e.g. De Kauwe et al., 2014). Yet with below-ground turnover ranging from 6 to 37% of NPP among models in the baseline simulations of the present study, addressing uncertainty related to variation in root exudates under environmental change is likely to remain a lower priority for modellers (Fig. 3).

**4.2 The role of phenology versus mortality in driving spatial variation in $\tau$ (H2)**

Much discussion has recently been devoted to potential changes in tree mortality rates and the resultant carbon cycle implications (e.g. Adams et al., 2010; Anderegg et al., 2012; Bennett et al., 2015; McDowell et al., 2018). Whilst the results of this study support the importance of mortality rates on determining $\tau$, they also demonstrate that different strategies in allocation to soft tissues are behind much of the spatial variation in $\tau$ in contemporary TBMs. In TBMs, phenological (and often mortality) turnover rates are strongly tied to PFTs (e.g. Table S2), reflecting different functional strategies, making simulation of the correct PFT distribution crucial to accurately determine $\tau$.

Furthermore, it is not clear whether the prevailing PFT paradigm, based largely on leaf phenology and type, appropriately captures the wider range of plant life-history strategies, which affect allocation of NPP and vulnerability to mortality, in trees in any given forest type (Reich, 2014; Salguero-Gómez et al., 2016). However, some TBMs, including LPJ-GUESS and SEIB-DGVM in the present study, do explicitly represent PFTs with contrasting life-history strategies, which may coexist in a stand

and affect the development of that stand (e.g. Hickler et al., 2004). Large trait databases (e.g. TRY; Kattge et al., 2011) and inventory datasets (Brienen et al., 2015; Hember et al., 2016; Ruiz-Benito et al., 2016) are being leveraged to better inform the

range of plant strategies employed (e.g. Christoffersen et al., 2016; Díaz et al., 2016; Liu et al., 2019; Mencuccini et al., 2019) and diversification of the strategies represented in TBMs, either through additional PFTs or flexible trait approaches (Langan et al., 2017; Pavlick et al., 2013; Sakschewski et al., 2015; Scheiter et al., 2013), may be necessary.

New cross-walking techniques (Poulter et al., 2015) help to resolve the inconsistency between satellite landcover

classifications (e.g. ESA CCI; ESA, 2017) and PFTs simulated by TBMs, facilitating a standardised benchmarking process for PFT distributions. However, global tree, and thus PFT, distribution is an amalgamation of natural dynamics and forest management activities. As large-scale forest management information is lacking, TBMs often simulate only the effect of natural dynamics on forest properties. Accurately representing the effect of forest management across the globe, such as recently developed for Europe (McGrath et al., 2015), will be crucial to simulating current PFT distributions and other forest

properties for the right reasons. Combining satellite landcover with inventory data will better capture forest management practices along with finer details of PFT distributions that elude current landcover classifications (Schelhaas et al., 2018). Hyperspectral remote sensing may also help provide greater fidelity in identifying different PFTs where reliable inventories are lacking (Asner and Martin, 2016).

### 4.3 Woody biomass: Long turnover times or high C allocation? (H3)

Observations from tropical forest plots point towards $\tau_{mort}$ being underestimated in the TBMs of this study (Fig. 6) and suggest that an over-allocation to wood in the tropics might be, to varying degrees, a common feature of TBMs. Because the carbon allocated to wood in TBMs is a trade-off with respiration and soft-tissue demands, this indicates that the latter might be underestimated. However, since increases in LAI or fine root density provide a diminishing return in terms of resource acquisition, understanding allocation to reproduction and defence may be the key to balancing tree carbon budgets. Efforts

described in Section 4.1 will greatly assist in closing this knowledge gap regarding allocation. However, H3 can be directly tested by strongly constraining $\tau_{mort}$ across all forests. The necessary information exists in forest inventory and research plot data for all major forest types (Brienen et al., 2015; Carnicer et al., 2011; Hember et al., 2016; Holzwarth et al., 2013; Lines et al., 2010; van Mantgem et al., 2009; Peng et al., 2011; Phillips et al., 2010), but this information needs to be collated and standardised such that consistent comparisons across regions can be made. A comprehensive database based on such data

could be used to benchmark TBMs by biomass turnover and, for individual or cohort models, stem turnover. Where possible, branch turnover flux, currently ignored in most TBMs, should also be assessed. If recently reported fluxes approaching 50% of woody turnover (Marvin and Asner, 2016) are widespread and broadly supported, the implications would propagate through the simulation of allocation and forest structure.

## 4.4 Processes causing tree mortality (H4)

To support accurate predictions in the context of global environmental change, mortality representations in models must reflect confirmed mechanisms and responses, resolving the very different hypotheses regarding the dominant form of tree death (Fig. 7). Fundamental to this effort will be including process information at a level of complexity appropriate for the scale to be simulated, and supportable by available data across biomes, stand types, and environments globally. For instance, it may not be possible to simulate explicitly the dynamics of a particular pest known to cause tree death in the absence of sufficient

quantitative data. But if the resulting mortality is closely associated with trees' ability to defend themselves in a given resource environment, a simplified or aggregate parameterisation linked to a metric of tree vitality such as 'growth efficiency' may provide an adequate substitute. The TBMs considered in this study combine a variety of mortality processes, which often bear a clear conceptual relation to observed drivers of tree death (e.g. low vitality, large-scale disturbance, maximum age/height). That they yield such different projections (Figs. 8, S9) is a result of challenges in both model parameterisation and

conceptualisation. Forest inventories and research plots may not provide insight into the proximate cause of death, but, assuming that woody growth is a good proxy for vitality (as in e.g. Schumacher et al., 2006), many inventory protocols give enough information to constrain the vitality and background processes outlined in Table 3. A first step is thus for modellers to further leverage the available data to adapt and better constrain existing approaches to simulating tree mortality.

Fully resolving H4 is likely to require inclusion of additional processes in TBMs, particularly the explicit representation of large disturbances and plant hydraulic failure. Whilst tree mortality from fire is explicitly included in many current TBMs (e.g. Table 3), tree mortality from ephemeral insect and pathogen outbreaks, which, at least in some regions, might be similar in magnitude to tree mortality from fire (Kautz et al., 2018) and liable to intensify with global warming (Seidl et al., 2017), is not to our knowledge part of any operational global model. Stand-replacing windthrow events, which are the main natural

disturbance in parts of temperate and tropical forests (Negrón-Juárez et al., 2018; Seidl et al., 2014), are another example of a key process missing in current models (but see Chen et al., 2018). Accounting for such disturbances through a process-oriented modelling approach (Chen et al., 2018; Dietze and Matthes, 2014; Huang et al., 2019; Landry et al., 2016) remains highly challenging in the absence of sufficient quantitative data on cause and effect. However, using prescribed, spatially, and where possible temporally, explicit disturbance fraction maps based on observations will help to improve simulations of carbon

turnover dynamics in current forests (Kautz et al., 2018; Pugh et al., 2019a). A first such map now exists for biotic disturbance for the northern hemisphere (Kautz et al., 2017), but the underlying data are scarce in many regions. For windthrow, probability maps do not currently exist at the global scale, but new generations of remote sensing products, building on the forest loss maps of Hansen et al. (2013), offer hope that this information will gradually become available in the coming years (e.g. Curtis et al., 2018; McDowell et al., 2015). Maximising the benefit from including such disturbances will, however, require TBMs

to explicitly track forest stand age, and indeed tree ages or sizes. TBMs which lump age/size classes will miss lagged sources or sinks resulting from how temporal changes in disturbances rates affect forest demography (Pugh et al., 2019b).

Lastly, much recent research has centred on the cause of death during drought, whether this is hydraulic failure, carbon starvation, phloem transport failure, or secondary biotic attack as a shortage of carbohydrate reduces the ability of the tree to
defend itself (Hartmann, 2015; Hartmann et al., 2018; McDowell et al., 2008; McDowell, 2011; Sevanto et al., 2014). Whilst vitality could provide an adequate proxy for most of these factors, hydraulic failure of the xylem transport system is conceptually distinct and the latest evidence suggests that it plays a major role in many ecosystems (Anderegg et al., 2015, 2016; Hartmann, 2015; Liu et al., 2017; Rowland et al., 2015). It is especially relevant to $\tau_{mort}$ because hydraulic failure appears more likely to occur in larger trees (Bennett et al., 2015; Rowland et al., 2015; Ryan et al., 2006), which hold a disproportionate
share of biomass carbon stocks, and whose death will create large canopy gaps for regeneration. There is currently no representation of hydraulic failure incorporated within the TBMs of this study; however, several efforts to achieve this are ongoing within the community (e.g. Eller et al., 2019; Kennedy et al., 2019; Xu et al., 2016). Large-scale evaluation of these representations will benefit from compilations of drought mortality events with increased event meta-data on cause of death, scale of the event and mortality rates (e.g. Greenwood et al., 2017), alongside exact locations and site characteristics such as
slope and soil type. Such meta-data will help to minimise scale mismatches and better resolve contributory factors.

## 4.5 Response of $\tau$ to environmental change: PFT establishment rates (H5)

Changes in $\tau$ over the 21st century will result from a combination of changes in mortality rates of existing trees and from a gradual establishment-driven shift in functional composition towards plants with different characteristic mortality or phenological turnover rates that better suit the new environment (Salguero-Gómez et al., 2016). Such compositional shifts
have been detected in the Amazon region (Esquivel-Muelbert et al., 2019) and in other taxa in Europe (Bowler et al., 2017). The TBMs used here display both behaviours. A shift in mortality rate of existing trees may also accelerate a compositional shift, seen here clearly in LPJmL for the boreal region, leading to a compound effect on turnover time, or it may leave functional composition largely unchanged. Better understanding of tree mortality processes and thresholds (see Section 4.4) will help identify the likelihood of alterations in mortality rate and the extent to which changes in mortality rates can occur without
triggering a shift in vegetation composition. However, accurately simulating establishment is clearly fundamental to assessing the long-term response. Establishment in TBMs is generally based either on NPP or the abundance of mature trees, often within defined bioclimatic limits (Krinner et al., 2005; Sato et al., 2007; Sitch et al., 2003). These representations may be too simple because they exclude three important factors. First, existing climatic relationships for establishment may not hold under elevated atmospheric $CO_2$ concentrations because of alterations in seedling assimilation rates (Hattenschwiler and Korner,
2000; Würth et al., 1998). This situation may require additional experimental work in chambers or plots with perturbed conditions such as FACE (e.g. Norby et al., 2016) to determine whether a change in seedling assimilation rates is likely to lead to a vegetation composition shift, thus affecting $\tau$ via MP mechanisms. Second, recruitment of new tree cohorts is strongly affected by the light and moisture environment at the forest floor (Muscolo et al., 2014; Poorter et al., 2019). Changing

mortality rates and driving mechanisms will affect canopy gap sizes, gap formation rates, and the intensity of the gap-forming disturbance (i.e. is the understory also lost?) (Beckage et al., 2008), influencing the ratio of early successional to late successional trees, which is highly likely to affect $\tau_{mort}$ (MP mechanisms in Table 1). Thus, representations of forest demography and canopy gap dynamics may be necessary in order to prognostically simulate establishment under changing environmental conditions. Third, seed dispersal limits the speed at which species composition changes in response to changing environmental conditions, with many plant species poorly predisposed to keep up with climate change (Corlett and Westcott, 2013) and some already lagging behind the spatial shift in their climatic niche (e.g. Zhu et al., 2012). Furthermore, not all species have the same dispersal abilities, with early successional species having on average higher dispersal abilities than mid- and late-successional species (Meier et al., 2012). Considering these three factors may substantially increase TBM complexity, therefore exploratory work is needed to more thoroughly assess their potential importance and to further develop parsimonious and scale-appropriate algorithms which focus on the most influential components of these processes. Some such developments are ongoing, e.g. in LPJ-GUESS (Lehsten et al., 2019).

### 4.6 Impact of elevated atmospheric $CO_2$ concentration on mortality (H6)

Reduced rates of mortality due to elevated atmospheric $CO_2$ concentration (H6b) are conceptually included in five of the TBMs through the growth efficiency concept (Table 3) and is evident in the overall response for two of them (Table 5). Increased plant production under elevated $CO_2$ follows well-established leaf-level responses of photosynthesis and water-use efficiency to atmospheric $CO_2$ concentration, and is supported by detailed stand-level modelling (Liu et al., 2017), but is hard to verify with observations in mature trees (Jiang et al., 2020; Walker et al., 2019). If trees expend their extra NPP on growing proportionally larger, thereby increasing their respiration demands, then the positive effect of enhanced NPP could be offset. Increased water-use efficiency under elevated $CO_2$ could also reduce mortality due to hydraulic failure (Liu et al., 2017), but none of the models considered in this study represent that interaction (Section 4.4).

Increases in NPP are also linked to mortality through competition (Table 1; $MS_{comp}$). Higher growth rates will increase the rate of vitality-induced mortality in forest stands (Pretzsch et al., 2014), thus acting to reduce $\tau_{mort}$. These relationships of tree size to stand density are very well established (Coomes and Allen, 2007; Enquist et al., 2009; Pretzsch, 2006; Westoby, 1984) and the process is included either directly, or via growth efficiency, in all of the TBMs considered (Table 3). This "self-thinning" process does not put a firm limit on stand biomass, as tree allometry means that large trees can hold more biomass than a larger number of smaller trees covering the same area. However, it means that reductions in tree mortality rates during drought extremes due to increased vitality resulting from increased atmospheric $CO_2$ concentrations will be at least partially offset by increased mortality rates through stand dynamics if extra NPP is invested in growth. Where the balance lies will depend on the frequency and severity of drought events, the level of competition between individual trees for resources and the slope of the density versus size relationship, which is known to vary across different forest types and with stand age (Enquist et al., 2009;

Pillet et al., 2018; Pretzsch, 2006). More extensive use of information from plot networks (e.g. Crowther et al., 2015; Liang et al., 2016; Brienen et al., 2015) could provide a relatively tight constraint on baseline mortality rates resulting from competition. Further, such data can be used for routine benchmarking of stand-level stem density vs biomass relationships in cohort and individual-based TBMs (Wolf et al., 2011).

## 4.7 Allocation of extra resources: Wood or elsewhere? (H7)

Given the lack of constraint regarding allocation fractions under current conditions (H1, Section 4.1), it is perhaps not surprising that the TBMs show different responses of allocation to increased productivity following $MI_{NPP,F}$ or $MI_{NPP,FS}$. Both hypotheses H7a and H7b are eminently plausible. If light and water/nutrient capture are already maximised then there is little advantage in further investment in leaves or fine roots, suggesting that allocation to these tissues should reach an effective limit. But, as with H3, whether the additional carbon is allocated preferentially to wood growth, or to rapid turnover items such as defence compounds, reproduction or exudates is unclear. Careful tracking of carbon in $CO_2$ enrichment experiments such as FACE will give answers for some ecosystems (Jiang et al., 2020; Norby et al., 2016) and can be used to set initial bounds on behaviour. Model parameterisation across a broader range of ecosystems may require setting these experimental outcomes in the context of how productivity and allocation vary in observations of individual tree species across resource gradients (e.g. Tomlinson et al., 2012), or relating allocation strategies to genetic drivers (Blumstein et al., 2018). This is an extremely challenging aspect of TBM behaviour to constrain, but the assumption made has a substantial influence on $\tau$ and biomass stocks in future climate simulations and should at least be clearly stated.

## 5 Conclusion

Biomass carbon turnover time is a high-level metric that integrates over a wide variety of underlying processes. Baseline turnover times at the global scale are highly uncertain and this uncertainty is caused not just by mortality, but also by a range of mechanisms that affect allocation to, and turnover rates of, soft tissues. A focus primarily on $\tau_{mort}$, on the grounds that most of the biomass is held within the wood of trees, is necessarily a static view of forests. In reality, forests are dynamic, their species composition and the allocation of carbon between different biomass compartments responding to changes in their environment, as reflected by TBM structures. Thus, constraining the current large uncertainty in overall woody carbon turnover rates is crucial, but so too is accurately assessing the conditions which favour establishment of individual tree types following mortality events, and quantifying for these individual tree types the characteristic mortality, allocation between wood and soft tissues, the turnover rates of these soft tissues, and how all of this varies among biomes, stand types and with the microenvironment of the tree.

It was not possible here to draw robust conclusions from the TBM simulations regarding likely variations in $\tau$ in different biomes or under the future climate compared to present day. Broadly, the mechanisms represented in different TBMs are

plausible given the state of current knowledge. Testing the identified model-based hypotheses will help to reduce both spatial and temporal uncertainty in $\tau$. Although testing some of these hypotheses will be challenging and require new observations, significant progress can be made using existing knowledge and data, particularly for H2, H3, H4 and H6a (Table 5). Key to this effort will be ensuring a smooth interface between TBMs and observations. This task requires efforts both to (1) compile and analyse observational data in ways that directly inform TBMs and (2) design or modify TBMs to ensure that they are structurally capable of using those data. For instance, accurately representing forest demography in TBMs is clearly central to simulating many of the important processes highlighted above, but it also allows the TBM simulations to be directly compared to, and constrained by, inventory data (Fisher et al., 2018; Smith et al., 2014). In some cases, confidence in TBMs may increase if they can simulate properties that are widely observed and can be used for constraining model simulations, such as satellite reflectance values. It will be important to incorporate observational data compilations into standardised benchmarking methods (e.g. Schaphoff et al., 2018). This benchmarking must go beyond the emergent property of turnover time, to the underlying processes, facilitating model improvement as well as evaluation. Rather than painting a dispiriting picture, the divergence of TBM estimates of $\tau$ reflects the ingenuity of scientists in the relatively data-poor world in which most TBM vegetation dynamics schemes were first developed. With the enormous increase in observational data over the last two decades, there is great potential for improvements.

**Data and code availability**

The model simulations described in this study can be accessed at https://zenodo.org/communities/vegc-turnover-comp/. Code for the analysis and figures in this study can be downloaded from https://github.com/pughtam/turnover_comp.git.

**Author contribution**

TAMP conceived the analysis. TAMP, JB, BB, VH, AH, JH, KN, AR and HS contributed model simulations. TAMP, TR, SLS, and JS carried out the data analysis and all authors contributed to data interpretation. TAMP, AA, SH, MK, BQ, TR, SLS, BS, KT and TH wrote the manuscript. All authors commented on the manuscript.

**Competing Interests**

The authors declare that they have no conflict of interest.

**Acknowledgements**

The research in this article resulted from the workshop *Dynamic Global Vegetation Modelling: Towards a third generation*, held in Landskrona, Sweden, in May 2015, and funded by the Swedish Research Council FORMAS through a grant for the project Land Use Today and Tomorrow (Dnr. 211-2009-1682), and by the Strategic Research Areas BECC and MERGE. T.A.M.P. acknowledges support from the European Research Council under the European Union Horizon 2020 programme (Grant 758873, TreeMort). T.A.M.P. and A.A. acknowledge support from European Union FP7 Grant LUC4C (Grant 603542), and the Helmholtz Association in its ATMO programme. J.B. was supported by the Centre National de la Recherche Scientifique (CNRS) of France through the program "Make Our Planet Great Again". S.L.S. was supported by the U.S. Geological Survey Land Change Science Program. This is paper number 42 of the Birmingham Institute of Forest Research. David Galbraith is thanked for providing turnover times from tropical plots. Jon Sadler and Thomas Matthews are thanked for advice on analysis, along with Dominique Bachelet and Todd Hawbaker for their comments on an earlier version of the paper. Any use of trade, firm, or product names is for descriptive purposes only and does not imply endorsement by the U.S. Government.

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

**Table 1. Conceptualisation of mechanisms by which biomass stock or τ can be modified as a result of environmental change. Many of these mechanisms may respond in concert to a given driver. Mechanisms are grouped by those related to the existing functional composition of trees and those related to a change of tree functional composition. The change in woody biomass and τ due to a change in NPP, resource allocation, mortality turnover rate or phenological turnover rate is illustrated. A dash indicates no change. Examples are only illustrative; the same mechanism could result from many scenarios and the listed examples may also influence other mechanisms. Further, the change for each mechanism is conceptualised in a particular direction, consistent with the given example, but could equally apply in reverse. For instance, MI_MR could also be shown with a decreased mortality rate, leading to increased biomass and τ. The groupings correspond to those commonly used in TBMs, with "mortality" referring to turnover from wood resulting from tree death, and "phenological" referring to turnover of "soft" tissues, which include leaves, fine roots and fruits. For simplicity, rapidly turned-over components such as root exudates and biogenic volatile organic compound emissions, which are rarely explicitly represented in TBMs, are lumped into the categories "soft" and "phenological" for allocation and turnover, respectively, although it is noted that some TBM parameterisations may implicitly include the lost carbon in respiration fluxes. Codes (e.g. MI_MR) are introduced and used in the main text to refer to the individual mechanisms.**

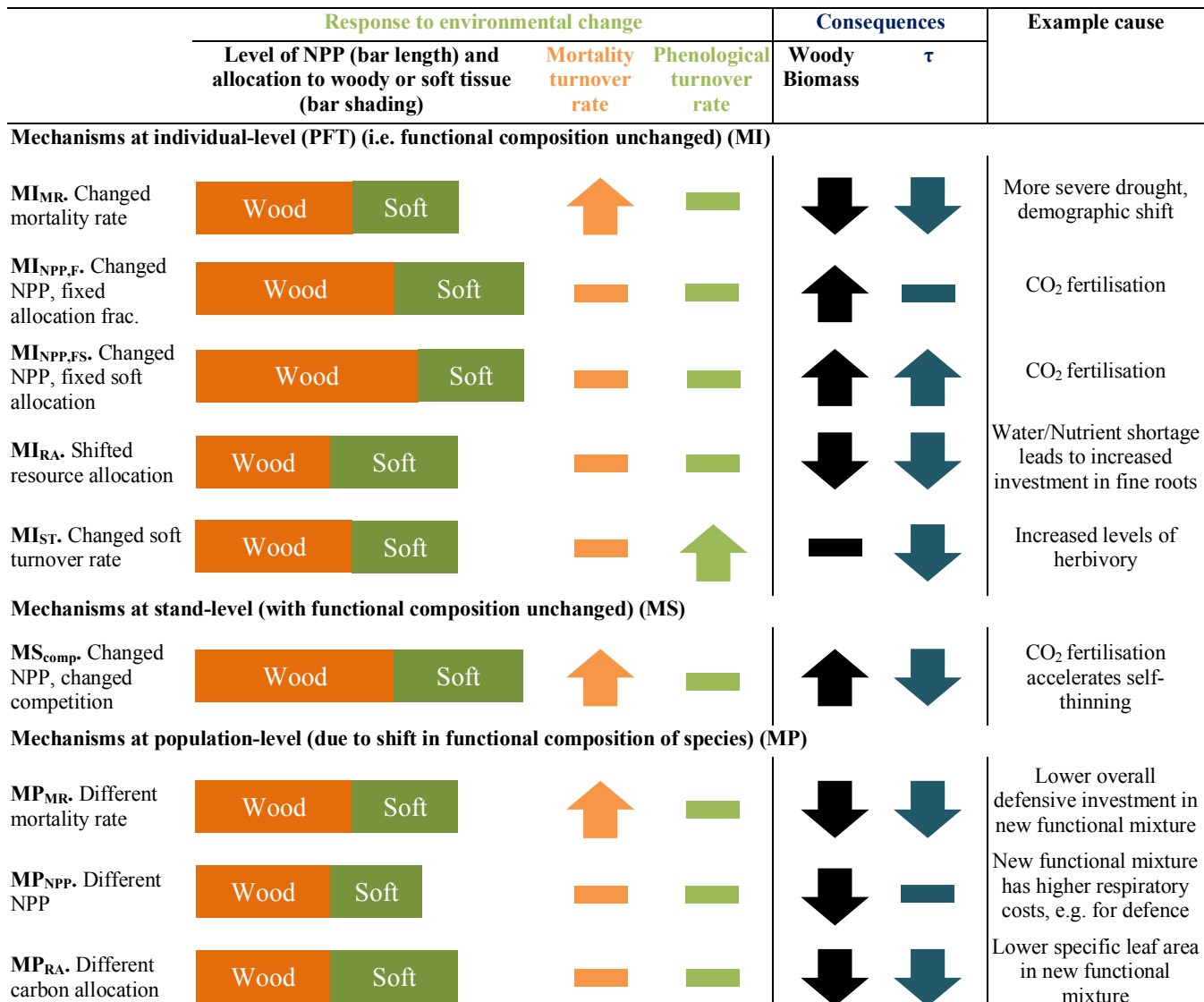

**MP_ST.** Different soft turnover rate

| Wood | Soft |

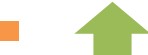 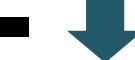

Shift towards deciduous leaf phenology

**Table 2. Models included in this study.**

| Model | Version | Dynamic vegetation | Vegetation representation | Key reference |
|---|---|---|---|---|
| CABLE-POP | rev. 4601 | No | Cohort | Haverd et al. (2018) |
| JULES | rev. 6679 | Yes | Average-individual | Clark et al. (2011) |
| LPJ-GUESS | rev. 4619 | Yes | Cohort | Smith et al. (2014) |
| LPJmL3.5 | rev. 3018 | Yes | Average-individual | Sitch et al. (2003), Bondeau et al. (2007) |
| ORCHIDEE | rev. 3085 | Yes | Average-individual | Krinner et al. (2005) |
| SEIB-DGVM | ver. 2.70 | Yes | Individual | Sato et al. (2007) |

**Table 3. Individual mortality processes included in the terrestrial biosphere models (TBMs) in this ensemble.**

| Conceptual grouping | Process | Example formulation (for actual model formulations see references in Table 2) | Included in model? | | | | | |
|---|---|---|---|---|---|---|---|---|
| | | | CABLE-POP | JULES | LPJ-GUESS | LPJmL | ORCHIDEE | SEIB-DGVM |
| Vitality | Growth efficiency | $mort_{greff} = \dfrac{k1}{1 + k2(\Delta C/LA)}$ where $k1$ and $k2$ are coefficients, $\Delta C$ is the annual biomass increment and $LA$ is leaf area. $mort_{greff}$ is a fractional scalar, where 1 = 100% mortality. | X | | X | X | X | X |
| | Self-thinning | if $\sum_{PFT} A_{PFT} > A_{max}$, then mortality occurs to reduce $A_{PFT}$, where $A_{PFT}$ is the ground area covered by a particular PFT and $A_{max}$ is the maximum allowable area coverage for all PFTs in a grid-cell. | X | X | X | X | | |
| Disturbance | Disturbance | Random likelihood of stand destruction in any given year with a globally defined typical return time (e.g. 100 years) | X | | X | | | X |
| | Fire | Thonicke et al. (2001) process-based fire model | | | X | X | X | X |
| Background | Max age/size | Trunk width exceeds maximum value, or increasing with age. | | | X | | | X |
| | Fixed turnover | Fixed turnover time for wood biomass (applicable in models using average individuals only) | | X | | | X | |
| Heat | Heat | $mort_{heat} = max\left[1, \dfrac{\sum_d max\,(T_d - T_{mort}, 0)}{M_{full}}\right]$ where $T_d$ is daily mean temperature, $T_{mort}$ is a base temperature for mortality, and $M_{full}$ is a temperature sum for 100% mortality. $mort_{heat}$ is a fractional scalar, where 1 = 100% mortality. | | | | X[a] | X[a] | [b] |
| Other | Bioclimatic limits | Multi-annual means of temperature fall outside a PFT specific range. | | | X | X | X | X |
| | Negative biomass | Biomass in any vegetation compartment becomes negative (NPP is more negative than living biomass) | | | X | X | | |

[a] Only implemented for the boreal PFTs.
[b] The original formulation of SEIB-DGVM includes heat stress mortality, but this function is now commonly turned off, as it was in this study.

**Table 4. 1985-2014 global closed-canopy forest totals based on the CRU-NCEP-forced simulations and satellite-based methods.**

| Model | NPP (Pg C a$^{-1}$) | C$_{veg}$ (Pg C) | $\tau_{NPP}$ (years) | $\tau_{turn}$ (years) | $\tau_{mort}$ (years) | $\tau_{fineroot}$ (years) |
|---|---|---|---|---|---|---|
| CABLE-POP | 18.4 | 414.0 | 22.6 | 23.5 | 49.9 | 0.6 |
| JULES | 24.0 | 284.1 | 11.9 | 12.2 | 15.1 | 5.0 |
| LPJ-GUESS | 23.0 | 288.7 | 12.5 | 13.2 | 36.0 | 1.4 |
| LPJmL | 22.9 | 429.2 | 18.8 | 19.8 | 47.5 | 1.8 |
| ORCHIDEE | 31.8 | 432.0 | 13.6 | 14.2 | 26.1 | 1.7 |
| SEIB-DGVM | 29.9 | 421.0 | 14.1 | 14.7 | 30.1 | 1.7 |
| Satellite-based | 23.3[a] | 449.7[b] | 19.3[b] | N/A | N/A | N/A |

[a] NPP calculated over 2000-2012.

[b] Nominal base year in range 2000-2010.

**Table 5. Hypotheses resulting from the terrestrial biosphere models (TBMs) for controls on spatial and temporal variation in turnover time.**

| | Hypothesis | Mechanisms | Models exhibiting response |
|---|---|---|---|
| **Existing situation (baseline)** | | | |
| **H1a** | Investment in soft tissues is a relatively small fraction of NPP, implying relatively rapid turnover times for wood ($\tau_{mort}$). | N/A | JULES |
| **H1b** | Investment in soft tissues is a relatively large fraction of NPP, implying relatively long turnover times for wood ($\tau_{mort}$). | N/A | CABLE-POP, LPJ-GUESS, LPJmL, ORCHIDEE, SEIB-DGVM |
| **H2** | Variation in phenological turnover fluxes is as important as variation in mortality turnover fluxes, in driving spatial variation in $\tau$. | N/A | CABLE-POP, LPJ-GUESS, LPJmL, ORCHIDEE |
| **H3a** | Carbon turnover times in tropical evergreen forests are much longer than for other forests, driven by long turnover times for wood. | N/A | CABLE-POP, LPJmL |
| **H3b** | Carbon turnover times in tropical evergreen forests are much longer than for other forests, driven by greater relative allocation of NPP to wood. | N/A | CABLE-POP, JULES, LPJ-GUESS, ORCHIDEE, SEIB-DGVM |
| **H4a** | The main driver of mortality carbon turnover fluxes in global forests is physical disturbance. | N/A | CABLE-POP, LPJ-GUESS |
| **H4b** | The main driver of mortality carbon turnover fluxes in global forests is low vitality. | N/A | JULES, LPJmL, SEIB-DGVM |
| **Under environmental change** | | | |
| **H5a** | Environmental change leads to large changes in the mortality rates associated with PFTs, which dominate the change in $\tau$ over the 21st century. | $MI_{MR}$, $MI_{RA}$, $MI_{ST}$ | LPJmL[1] |
| **H5b** | Shifts in forest functional composition, rather than changes in the turnover rates associated with PFTs, dominate the response of $\tau$ to environmental change over the 21st century. | $MP_{MR}$, $MP_{RA}$, $MP_{ST}$ | LPJ-GUESS, JULES[1] |
| **H6a** | Elevated atmospheric $CO_2$ concentrations result in greater rates of mortality due to vitality-based processes because of increased competition for space as a result of increased NPP. | $MS_{comp}$ | CABLE-POP, JULES, LPJ-GUESS, SEIB-DGVM[1] |
| **H6b** | Elevated atmospheric $CO_2$ concentrations result in reduced rates of mortality because vitality-based processes are triggered less with increased NPP. | $MI_{MR}$ | LPJmL, CABLE-POP[1] |
| **H7a** | Increased forest productivity results in much higher relative allocation to wood than soft tissue, partially compensating for, or even outweighing, reductions in $\tau_{mort}$. | $MI_{NPP,FS}$ | JULES, LPJ-GUESS, LPJmL, SEIB-DGVM |
| **H7b** | Increased forest productivity has very little effect on relative allocation between wood and soft tissues. | $MI_{NPP,F}$ | CABLE-POP, ORCHIDEE |

[1] This hypothesis may hold in other TBMs here, although not positively identified in this study.

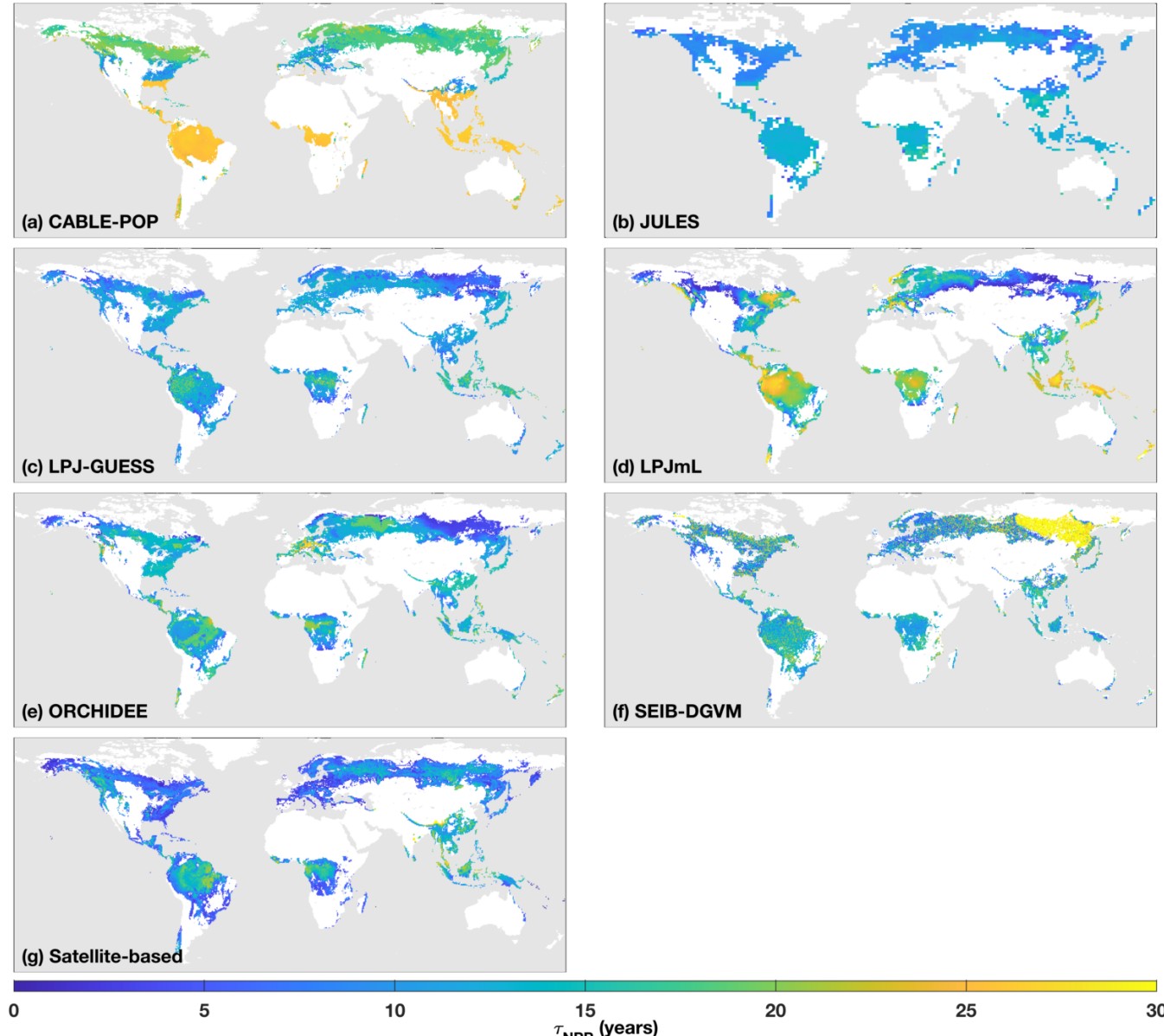

**Figure 1.** $\tau_{NPP}$ **mean for the period 1985-2014 as forced by the CRU-NCEP climate (units of years). Colour scale is capped at 30 years. Maps show areas which are simulated as forest for each model and have at least 10% of the grid-cell covered by closed-canopy forest based on Hansen et al. (2013) (see Methods).**

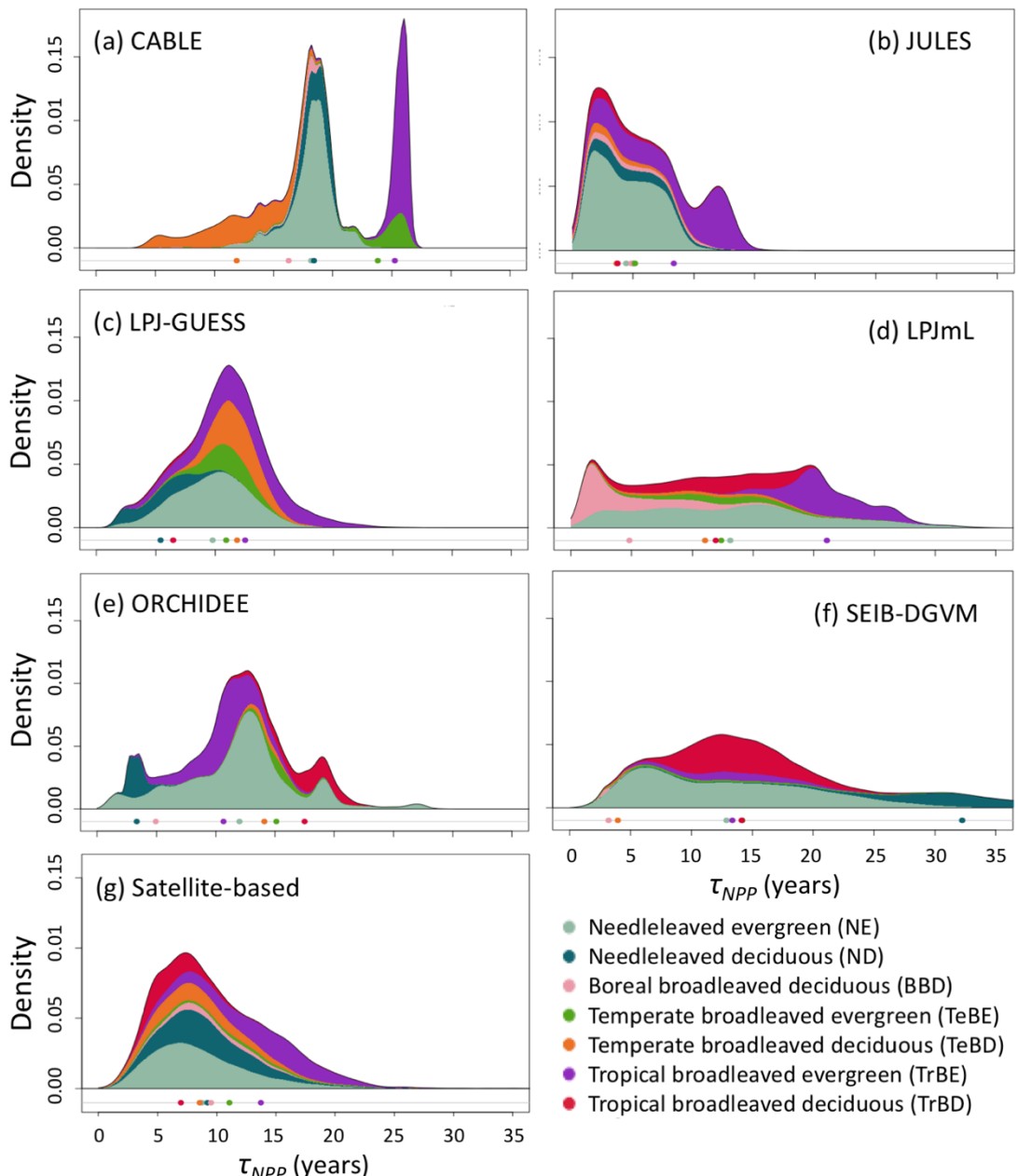

**Figure 2.** Probability density distributions for $\tau_{NPP}$ for the period 1985-2014 under CRU-NCEP climate calculated by forest type (see Methods) and superimposed to produce a global probability density distribution. Density is defined as fraction of total grid-cell number, including all grid-cells with at least 10% forest cover (i.e. masking as for Fig. 1). For the models, $\tau_{NPP}$ was derived from entire grid cell $C_{veg}$ and forested area-weighted NPP, as for the satellite-based product (see Section 2.4). Circles underneath distributions show the mean turnover time for each forest type after weighting by the forest cover fraction of the grid cell and excluding grid cells with less than 10% forest cover (see Table S1 for forest type definitions). For the satellite-based probability density distributions the observationally based forest types (Table S3) were used, with broadleaved-needleleaved mixed forest (MX) assigned to BBD and excluding other tropical forest (OTr) and other forest (Other) because no equivalent categories were reported for the models.

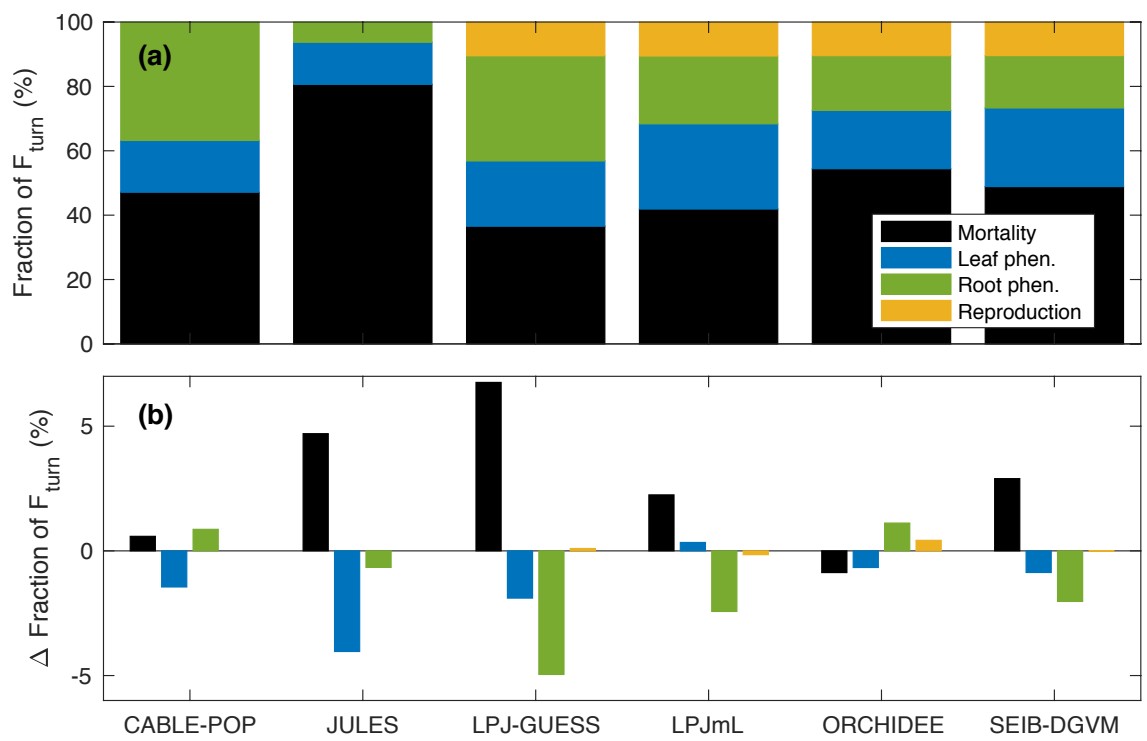

**Figure 3. Fraction of global F_turn resulting from individual model processes. (a) For 1985-2014 in the CRU-NCEP-forced simulation. (b) Change in fraction of F_turn (percentage points) between 1985-2014 and 2070-2099 in the simulations forced by IPSL-CM5A-LR RCP 8.5 bias-corrected climate data. Black is mortality, light blue is leaf phenological turnover, green is root phenological turnover, and yellow is reproductive turnover.**

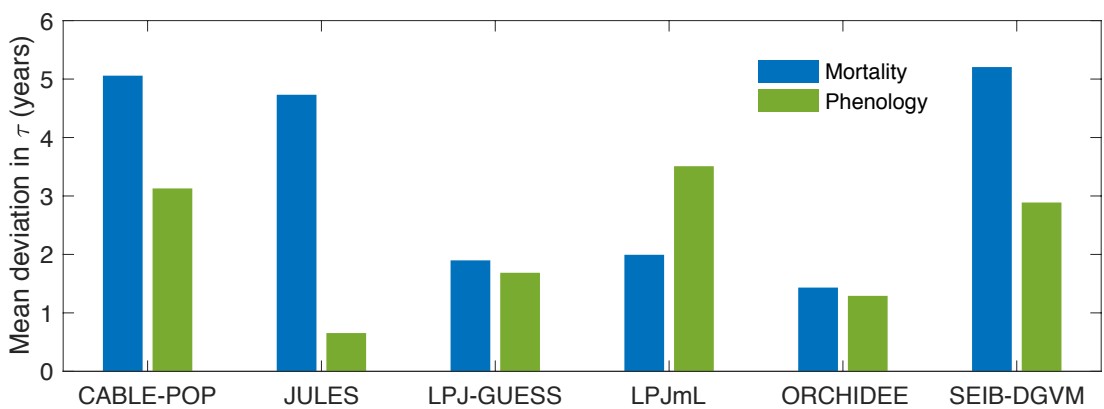

**Figure 4. Mean absolute deviation in $\tau_{turn}$ across all grid cells with at least 10% forest cover as a result of using global mean values of mortality ($F_{mort}$) or phenology ($F_{phen} = F_{leaf} + F_{fineroot} + F_{repro}$) turnover fluxes in the calculation of $F_{turn}$ in Eq. 2 (see Methods). Larger values indicate a greater contribution of $F_{mort}$ (blue) or $F_{phen}$ (green) to spatial variability in $\tau_{turn}$. Calculated over the period 1985-2014 from the CRU-NCEP-forced simulation.**

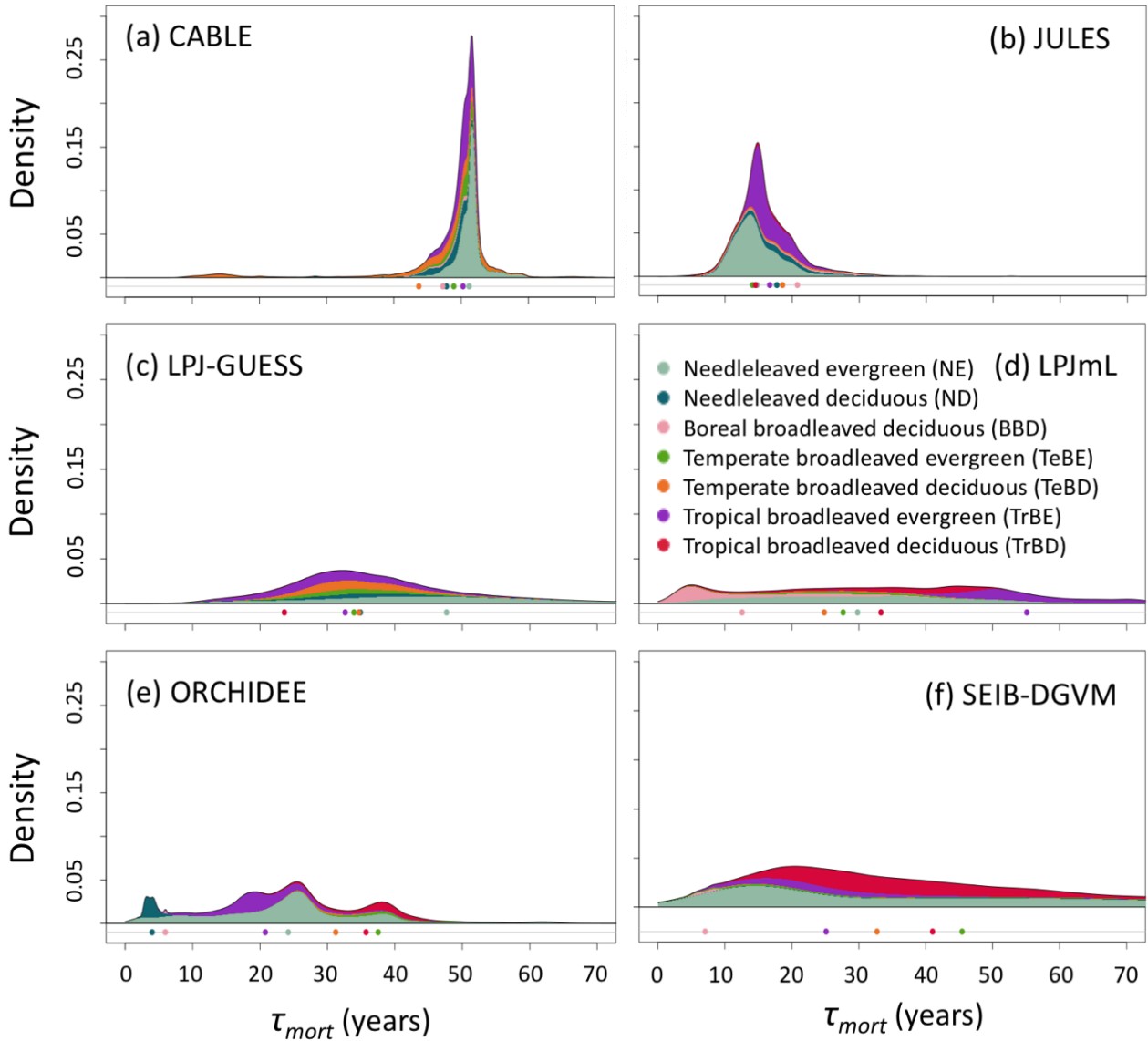

**Figure 5. As for Fig. 2, but for turnover times due to mortality alone, $\tau_{mort}$ ($C_{veg}/F_{mort}$). $\tau_{mort}$ was derived from entire grid cell $C_{veg}$ and forested area-weighted NPP, as for the satellite-based product (see Section 2.4) and grid cells were classified according to dominant PFT. Circles underneath distributions show the mean turnover time for each forest type.**

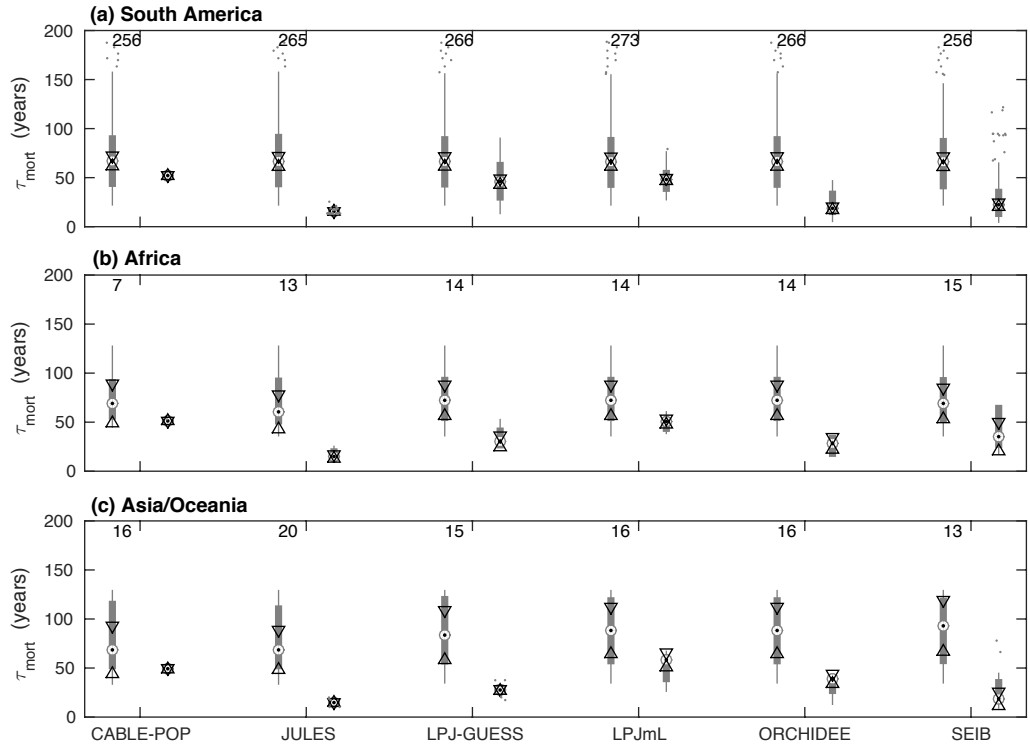

**Figure 6. Comparison of $\tau_{mort}$ from observations at forest plots across the tropics against modelled values of $\tau_{mort}$ obtained for the same sites. For each model, boxplots on the left show the observations and on the right the model results. Observations are shown separately for each model because some sites were not simulated as forest by some of the models. The number of sites included in the comparison is shown above the bars. Circles with dots show the median, with triangles identifying its 95% confidence limits. Thick grey bars show the interquartile range, with thin grey bars extending to the 10[th] and 90[th] percentiles. Outliers are marked with dots (horizontal spread illustrative only). The y-axis is truncated at 200 years.**

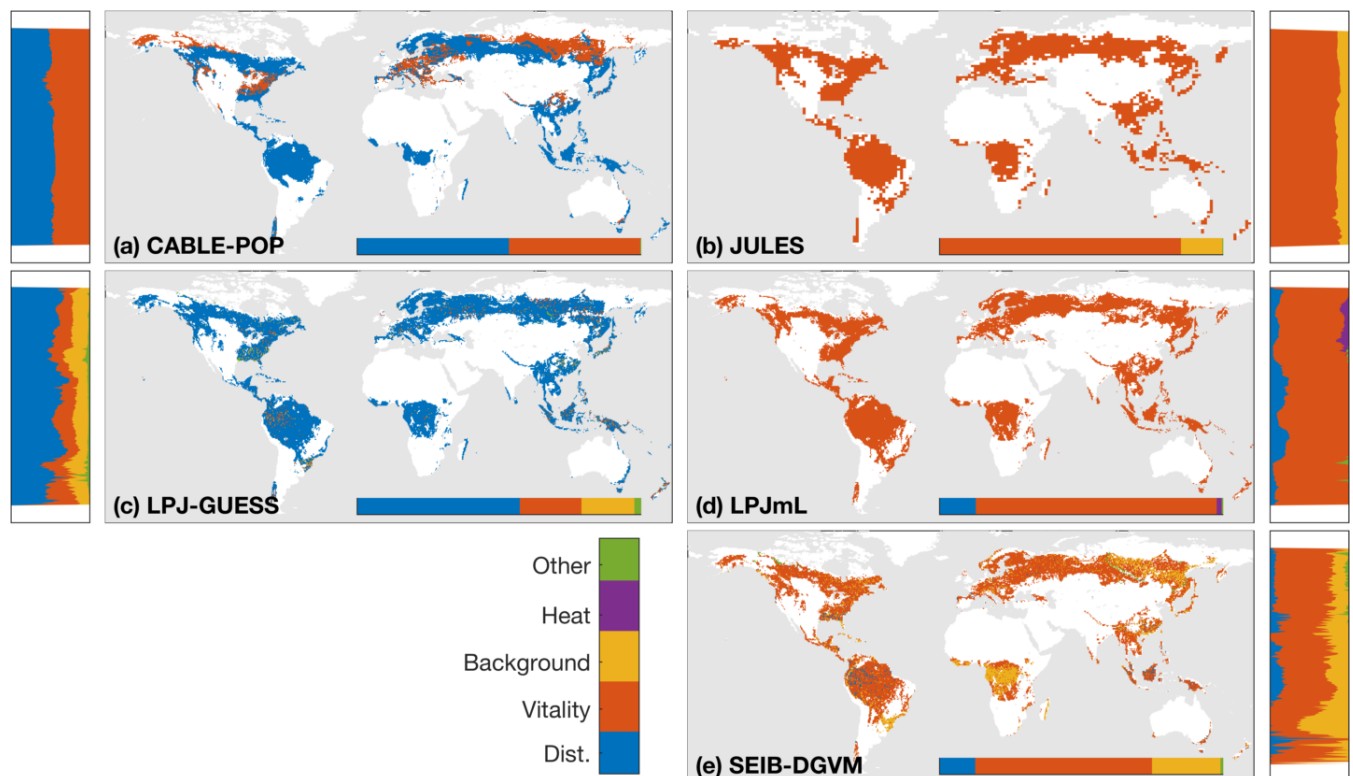

**Figure 7. Dominant mortality process by carbon flux for the period 1985-2014 as forced by the CRU-NCEP climate. Bar insets indicate the fraction of the global mortality-driven turnover flux due to each mechanism, whilst vertical side bars show the fraction due to each mortality process across latitude bands. Processes are grouped conceptually following Table 3 and equations and parameters used generally differ between models. "Dist." is mortality due to forest disturbance and may or may not conceptually include fire, depending on whether the model has an explicit fire mechanism. "Vitality" groups processes such as growth efficiency, self-thinning and more general competition. "Background" covers mortality based on a fixed rate or tree age. "Heat" is heat stress mortality. "Other" includes all processes that did not conceptually fit into one of the categories (Table 3). A breakdown of processes was not available for ORCHIDEE.**

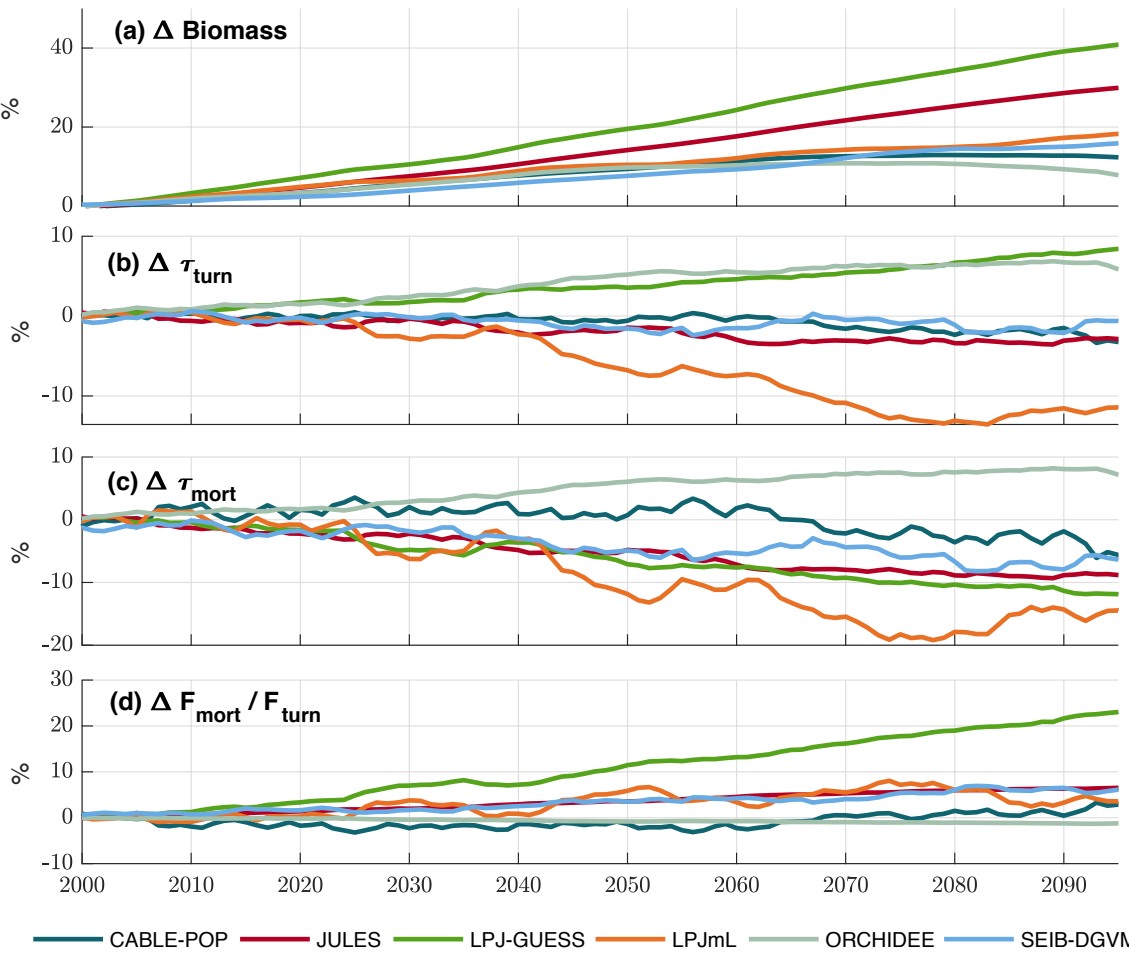

**Figure 8. Simulated evolution of carbon residence times in the TBM simulations forced by IPSL-CM5A-LR RCP 8.5 bias-corrected climate data. All plots show relative changes compared to a 1985-2014 baseline. (a) $C_{veg}$. (b) $\tau_{turn}$ ($C_{veg}/F_{turn}$). (c) $\tau_{mort}$ ($C_{veg}/F_{mort}$). (d) Change in fraction of total turnover due to mortality ($F_{mort}/F_{turn}$). Results are shown as an 11-year running mean.**

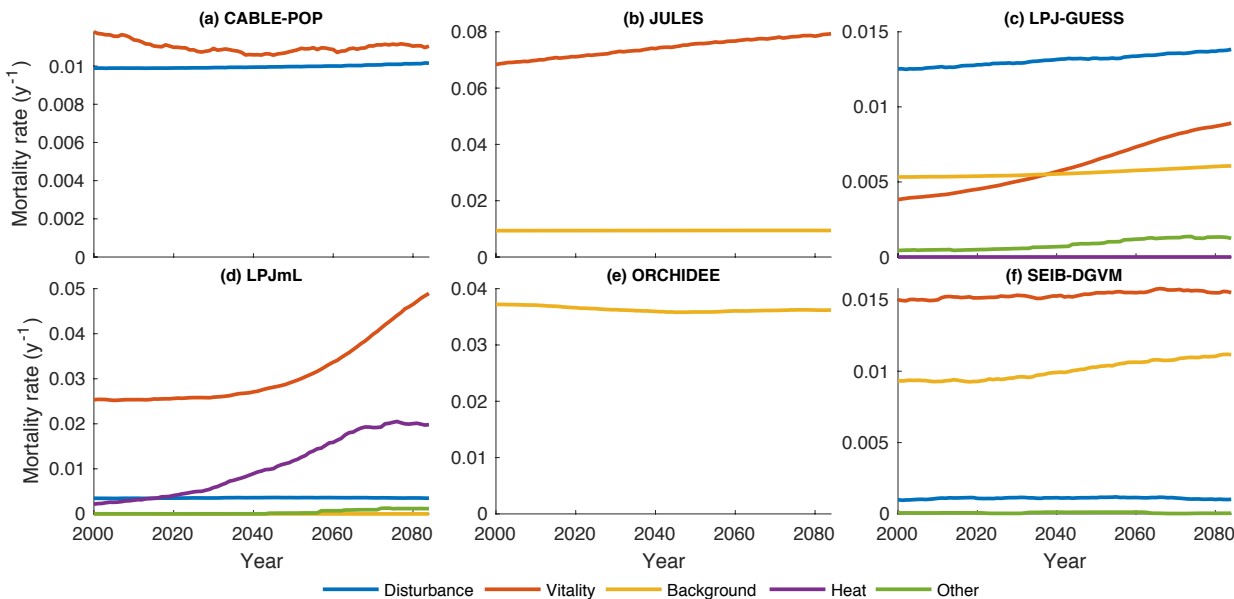

**Figure 9. Mortality rate ($F_{turn}/C_{veg}$) for the needleleaf evergreen forest type split by conceptual process grouping (Table 3) for the period 1985-2099 in the simulation forced by IPSL-CM5A-LR RCP 8.5 bias-corrected climate data. Observational forest types were used. 31-year running means are plotted for clarity and thus only 2000-2085 is shown. No process breakdown was available for ORCHIDEE, hence all processes were designated as "background". Note y-scales differ between panels.**

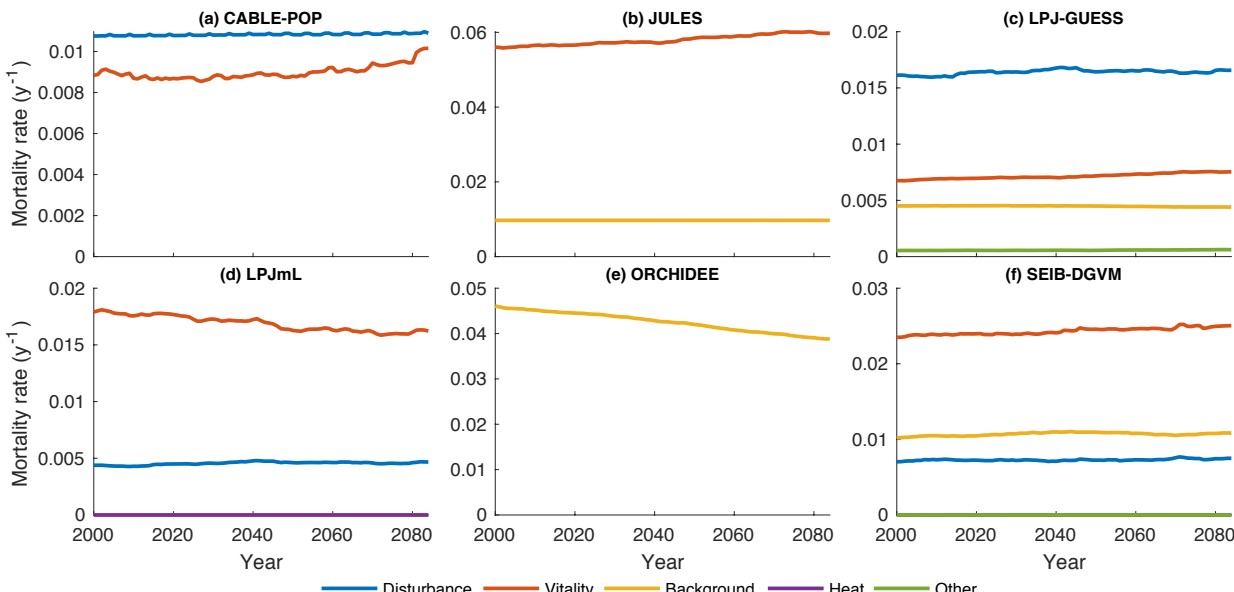

**Figure 10.** Mortality rate ($F_{turn}/C_{veg}$) for the tropical broadleaf forest type split by conceptual process grouping (Table 3) for the period 1985-2099 in the simulation forced by IPSL-CM5A-LR RCP 8.5 bias-corrected climate data. Observational forest types were used. 31-year running means are plotted for clarity and thus only 2000-2085 is shown. No process breakdown was available for ORCHIDEE, hence all processes were designated as "background". Note y-scales differ between panels.