# Peer review of "Understanding the uncertainty in global forest carbon turnover"

_Biogeosciences, 2019_

## Referee Comment (RC1) · Anonymous Referee #1 · 18 Mar 2020

**Review on**

**Thomas A. M. Pugh et al.**

**"Understanding the uncertainty in global forest carbon turnover"**
**submitted to Biogeosciences, Feb. 2020**

March 18, 2020

This paper analyzes turnover of forest carbon simulated by six TBMs (terrestrial biophere models) using a common simulation protocol "to assess their current capability to accurately estimate biomass carbon turnover times in forests and how these times are anticipated to change in the future." (abstract). The concentration on turnover is motivated by referring to studies that according to the authors have shown that turnover time has "comparable or even larger importance than NPP when assessing the response of $C_{veg}$ to environmental change" (lines 69/70). The analysis proceeds by first setting up a list of potential mechanisms that might modify turnover time upon environmntal change (table 5). Then, splitting the turnover fluxes obtained in the different simulations into components arising from different mechanisms (phenology, mortality), the analysis of the simulation data is condensed into a set of hypotheses on the controls for the behaviour of turnover time seen in the different models. The main result of this analysis is that in the different models the dominant mechanisms controlling turnover are different so that more research is needed. To direct such research the authors discuss for their hypotheses on the diverse controls how the scientific community could proceed by making use of existing data and what the challenges are in achieving progress.

Overall, I really don't know what to make out of this paper. The result from the analysis of the different turnover fluxes that in the different models the overall turnover is caused differently is not much surprising since this diversity is "strongly linked to model assumptions concerning plant functional type distributions and their controls" (lines 47/48), which are different across models. Making at all something out of this mess by coming up with a list of hypotheses on the different controls in the various models (summarized in the paper title somewhat overstated as "understanding the uncertainty"), is a rather heroic act, demonstrating the author's solid knowledge on ecological processes and their implementation into models. But then selling this mess by coming up with recommendations for the community on how to proceed makes the whole paper resembling an (extended) introduction for an application for funding rather than being a text informing the interested comunity on advances in understanding. I do not deny that this type of reasearch must be done to advance – and I appreciate the author's sophistication in their analysis – but it seems to me research that needs to be done *before* performing the true research because it is "only" formulating questions and setting up the research program. Accordingly, I am not convinced that such type of research that stops where it starts to get interesting is of sufficient interest for a larger community to be published in a reviewed journal instead of simply staying a paper in the grey literature.

Otherwise the paper is well readable, although the structure of the paper, outlined in the first paragraph above, could be emphasized stronger within the main text of the paper to keep the reader aware of the purpose of the different sections within the overall construction of the paper so that the 'red line' is better visible; e.g. only later I realized that the aim of section 3 (Results) is to identify the hypotheses listed in table 5 that subsequently, in the discussion part (section 4), are further discussed.

One other general issue concerns the specification of author contributions (Lines 605-7): Looking through the list, 3-4 authors remain that only commented on the manuscript without indication of any significant contribution to it. According to good scientific practice, this is not sufficient to justify authorship (otherwise I as reviewer should also be listed as author): *The European Code of Conduct for Research Integrity* [1] specifies "...authorship ...is based on a significant contribution to the design of the research, relevant data collection, or the analysis or interpretation of the results.". Hence for this paper authorship should be reconsidered. (And btw: who is "KZ"?)

In addition to this, I list below a number of remarks concerning the scientific contents of the paper and their presentation.

**Major remarks**

**(1)** Underlying this study is the general picture that vegetation carbon is growing due to $NPP$ and diminishing due to "turnover" fluxes called collectively $F_{turn}$. This is embodied in the first part of eq. 1, the rate equation $dC/dt = NPP - F_{turn}$. During the analysis presented in the paper, $F_{turn}$ is further subdivided into fluxes arising from different processes. Hence underlying this analysis is also a formula like $F_{turn} = F_1 + F_2 + F_3 + \ldots$. Moreover, for each such partial turnover flux a separate characteristic time is defined by a formula like $\tau_i := C_{veg}/F_i$. All this structure is not made explicit and transpires only while reading. I think for a better readability it would be helpful to make this structure explicit right at the beginning, maybe directly in connection with eq. 1. This would also be the right place to explain in detail the processes underlying all the partial turnover fluxes distinguished in this study, because these explanations are scattered across the paper (what is eg. "Background" in Fig. 6?).

**(2)** Furthermore, for the analysis framework it is important to note that all partial timescales are defined with reference to the same carbon mass $C_{veg}$. By this common reference the flux decomposition can also be written as $F_{turn} = C_{veg}(\tau_1^{-1} + \tau_2^{-1} + \tau_2^{-1} + \ldots)$ so that the rates $1/\tau_i$ indicate the size of the relative contributions of the different turnover processes to the total turnover flux. Since it is hardly believable that for all the different turnover processes the fluxes are proportional to the total biomass $C_{veg}$, the meaning of those time scales $\tau_i$ is only diagnostic and may have nothing to do with the actual time scales underlying the various processes causing turnover. An example is fires that depend

on the amount of fuel (litter or crown biomass) but not on stem biomass that contains the majority of vegetation biomass. This poses a serious question mark on the overall approach taken in this study by putting time scales into the center of the analysis. – Nevertheless, ratios of the diagnostic time scales stay well interpretable as ratios between turnover fluxes because in this framework $\tau_i^{-1}/\tau_{turn}^{-1} = F_i/F_{turn}$.

(3) For the analysis done in this paper the authors hop between comparing fluxes $F_i$ (e.g. Figs. 6 and 7d) characteristic times $\tau_i$ (Table 4, Fig. 7), rates $\tau_i^{-1}$ (Figs. S11-S15), and the additional time scale $\tau_{NPP} := C_{veg}/NPP$ (Fig. 1). It would be good to have a discussion on the advantages of using the one or other quantity. Anyway it should be made transparent why $F_i$, $\tau_i$, $\tau_i^{-1}$, or $\tau_{NPP}$ are chosen in the different parts of the analysis and why not the respective others.

(4) The descriptions in the main text heavily draw on plots in the supplement and there seems to be no systematics in putting plots here or there. E.g. for all the many claims on p. 11 the reader is referred to plots in the supplement which is in my opinion a good reason to move those plots – maybe in aggregated form – into the main text. And if the authors stick to their decision to have Fig. S9 in the supplement, then also the description on how to arrive at that figure (lines 221-229) should be shifted to the supplement. In conclusion, it seems to me that the readability of the paper could be improved by reconsidering the division of plots between main text and supplement.

(5) As already mentioned above, the study is motivated by the claim that $\tau$ has "comparable or even larger importance than NPP when assessing the response of $C_{veg}$ to environmental change" (Lines 69-73). Because this is the major justification of this study, the authors should be more precise here. E.g. Ahlström et al. (2015a) demonstrate that for stationary conditions turnover has only about 2/3 the importance of NPP (l.c. Fig. 3c), and should (to my understanding) be even less important for the non-stationary conditions of environmental change one is actually interested in. Even better would be to add further plots to this study demonstrating explicitly the author's claim by separating contributions to the biomass changes from NPP and turnover; an analysis in the spirit done by Friend et al. (2014) applied to the transient simulations in the paper would do it. To add: Simple reference to that Friend et al. paper will not be sufficient because their main message is so cryptically formulated that it is inconceivable (they write: "The variance in final $C_{veg}$ caused by differences in fitted residence time relationships between models was found to be 30% higher than that caused by differences in the fitted NPP responses when all models were considered." (p. 3283). What 'variance' is meant here, and why only "when all models are considered"?)

**Minor Remarks**

- Line 64: Here you introduce as key term the word 'residence time', nowhere else used in the paper, and the term 'turnover time' used throughout the paper, is only introduced in brackets. It's better to drop 'residence time' because its meaning is anyway ambiguous (see Sierra et al. 2017).

- Lines 75/76: For stationary states I agree that there are only these two causes, but for nonstationary conditions another one appears: Once NPP is changing, the internal memory causes a mismatch between NPP and $F_{turn}$ so that the turnover time $\tau_{turn}$, which is only a diagnostic obtained from $C_{veg}/F_{turn}$, changes in addition to the causes mentioned because the input NPP changes. I think the authors are aware of this (see lines 141/42 where the difference between $\tau_{NPP}$ and $\tau_{turn}$ is addressed), but since one is ultimately interested in the nonstationary case of environmental change this should not be overlooked.

- Throughout the paper the term "vitality" is used 28 times to refer to a certain group of processes. In lines 95/96 two of them are named, but there is no explanation on the general notion of this term. But such a general notion is needed to understand e.g. the claims in lines 97, 119, 295-298 etc. To me a formulation like "vitality-related mortality" (lines 320/21) sounds a bit weird: Increased mortality improves vitality? Without further help I have problems to make sense out of this vitality concept.

- I find it hard to memorize the meaning of your notation 'MI', 'MP', 'MS'. Maybe you indicate why you choose these combimations of letters or switch to a more mnemonic notation.

- Table 1: It took me really a hard time to understand how columns 2-4 should be read, the caption is in this respect rather unclear. Now I got that these are a pictorial representation of the assumptions on the topics in the headline. To prevent that other readers also struggle with this, I suggest to have another headline covering columns 2-4 eg. with title "assumptions on" and columns 5 and 6 with title "consequences for". Nevertheless, I still do not understand how to read column 2 "Resource capture (NPP) and allocation to woody or soft tissue": Has the combined length of the bars for "Soft" and "Wood" any meaning in relation to some maximum length, e.g. the largest combined bar or the width of the column? Or do you want to emphasize the relative lenghts of the bars "Soft" and "Wood"? Does a "Wood" bar shorter than the associated "Soft" bar mean that the influence on the former is smaller?

- Table 1: In some cases the consequences for biomass and $\tau$ shown in columns 5 and 6 seem to me rather speculative and do not follow directly from the information given in columns 1-4. Therfore this table should be revised or at least discussed in detail somewhere. My concerns on the proposed mechanisms are:
  $\mathbf{MI_{NPP,F}}$: Upon rising NPP it is claimed that $\tau$ remains unchanged. This is only true for very slow changes in NPP so that the system essentially stays in equilibrium. For a faster increase of NPP the turnover flux stays behind NPP because of the internal memory so that $\tau_{turn} := C_{veg}/F_{turn}$ changes even though only the input to the system changes. (See also comment on lines 75/76 above.)

**MI$_{RA}$**: A typical experience when experimenting with water stress in TBMs is that by increasing water stress, in dry regions annual production may increase. The reason is that the resulting reduced NPP flux leads to water savings so that the growth period is lengthened, to the consequence that the annual production (accumulated NPP) rises. For your table entry this would mean that you can as well reverse the direction of your biomass arrow. What the consequences for $\tau$ are would be hard to say.

**MI$_{ST}$**: The same remark on the counter-intuitive reaction on increased water stress applies also here so that biomass may increase. On the other hand: By increasing exudates it would also be plausible that biomass would decrease.

**MP$_{MR}$**: For the example driver "Reduced defensive investment" the consequences could be indeed as you describe if the investment was optimal or sub-optimal before. But in case of super-optimal investment in defense, biomass could as well rise so that also the converse behaviour of biomass and turnover rate is conceivable. And, just to note, it doesn't matter whether the reduced investment in defense arises from a shift in functional composition or not.

**MP$_{NPP}$**: I do not understand what is meant by "intrisic NPP" and in what respect the example driver is assumed to be "conservative".

**MP$_{RA}$**: I do not see why a shift to species with reduced wood density may necessarily decrease the characteristic turnover time: This is – as far as I see – only true when those species with reduced wood density have themselves lower than average turnover time. If not, turnover time may as well increase.

**MP$_{ST}$**: In the table you claim that upon an increased phenological turnover rate due to a shift in functional composition biomass stays unchanged. This seems to me rather unintuitive: This assumes that all species have the same productivity per leaf or root area ("effectivity"), which is typically not the case. Therefore, if the composition is shifted to more effective species the overall LAI or root area may decrease but the total productivity and therefore biomass may increase. And also the other direction is conceivable.

- Lines 186-189: It is unclear why this logic for identifying the forest mask was chosen. What goes wrong when one of the conditions (e.g. the condition on boreal PFT) would be omitted?
- Line 235: Wilkox**a**n $\rightarrow$ Wilkox**o**n.
- All figures: Put units at color scale.
- Figure 2: According to the caption the figure shows "density kernels". In the caption you explain "density", but the term "density kernel" is nowhere explained. I guess that you show the global relative abundance of $\tau_{NPP}$ values. Please consider a renaming or explain the term "density kernel" somewhere.
- Line 260: Here you claim that your findings are summarized in the alternative hypotheses H1a and H1b listed in table 5. But in the table entry you link the allocation fraction of NPP to soft tissues or wood to $\tau_{mort}$, but in the text leading to your claim you argue with the turnover fluxes and neither NPP nor $\tau_{mort}$ are mentioned, so that the link between text and hypotheses is missing. Moreover, I do not understand why there should be such a close link between the allocation fraction and $\tau_{mort}$ as you

claim in either of the two variants of the hypotheses H1 in the table text, it is easily imaginable that allocation fraction and $\tau_{mort}$ are completely unlinked.

- Lines 260/61, 314/15 and Fig. 4: At those lines you make clonclusions from Fig. 4 on the relative contributions of $F_{phen}$ and $F_{mort}$ to the variability of $F_{turn}$. I doubt that such conclusions can indeed be drawn from that figue. Such conclusions could be drawn if the fluxes $F_{phen}$ and $F_{mort}$ would be uncorrelated, but because $F_{phen} + F_{mort} = F_{turn}$ a large phenological flux leaves not much room for variation in the mortality flux and conversely. Therefore those two fluxes should be strongly (inversely) correlated and the condition of vanishing correlation to draw your conclusions is not met. Accordingly, I think this analysis should be discarded.
- Line 265: Where did you already follow "the same logic"?
- Line 295: What is the reason to start the sentence with "However"?
- Fig. 6: What is "Background"? In the caption: "vertical axis" $\rightarrow$ "vertical side bar" (e.g.).
- Fig. 7: I am not sure what you really show in subfigure d: In the caption you write that you show the "Fraction of total turnover due to mortality" but all curves start at zero so that some change is shown. Maybe you show the relative change in $F_{mort}/F_{turn} = \tau_{turn}/\tau_{mort}$? If so it would be more clear to write down this formula. I also don't know how to understand the in-figure text "$\Delta$ Fraction as mort." – why "as"?
- When distinguishing between PFTs it remains unclear whether characteristic times (Figs. 2, 5) and rates (Figs. S11-S15) are calculated with reference to total $C_{veg}$ obtained for the whole mix of PFTs in a grid cell, or with reference to the $C_{veg}$ of the individual PFTs in a grid cell. This difference has e.g. a large impact on whether changes displayed in Fig. 8 can be interpreted on the basis of Figs. S11-S15, as done on p. 11.
- Lines 339-41 and hypothesis H5/H6 in table 5: To me the description in lines 339-41 is inconsistent with the resulting formulation of hypotheses H5 and H6. In line 340 it is said that LPJmL's "increased mortality of established trees" is the reason for the shift in PFT composition – hence LPJmL cannot fall under H5, where "changes in turnover rate of individual PFTs" is explicitly excluded as cause, and the latter exclusion covers the former. Since LPJ-GUESS is listed to fall under H5 *and* H6 (which I do not understand) I guess that there is some insufficient distinctness in the formulations of the text and the hypotheses that make it impossible to follow what is meant here.
- Lines 347/48: Here it is claimed that JULES "implicitly" falls under hypothesis H5. But why is it then not listed in table 5 under H5?
- Lines 383-386: I guess "(1)" refers to MI and MS in table 1, while "(2)" refers to MP in that table. Its a bit irritating that you claim to have "identified" (line 383) these *two* groups "(1)" and "(2)" – in table 1 it were *three* groups and they were not "identified" but postulated. How did you "identify" these two goups?
- Lines 389-391: I have a problem to understand this sentence: What is meant by "These differences"? Differences because $MI_{ST}$ and $MP_{NPP}$ don't show up? "These" cannot refer to differences in the process diversity implemented into the models, because such

differences were not addressed before. Hence you mean differences between implemented processes and "emergent response"? This makes no sense. I am lost. And what means "have been under-constrained"? So they are constrained now?

- Lines 403-405: Wrong grammar: "...however ...which is usually absent ..."?
- Line 458: "that is key" $\rightarrow$ "is key"?
- Lines 563/64: The construction of the sentence is a bit weird. Maybe better: "...conditions (H1, Section 4.1), it is perhaps not surprising that the TBMs show different responses of allocation to increased productivity following mechanisms $\text{MI}_{NPP,F}$ or $\text{MI}_{NPP,FS}$.".

**Literature**

[1] *The European Code of Conduct for Research Integrity*, ALLEA - All European Academies, Berlin 2017, Revised Edition, `https://allea.org/code-of-conduct/`.

---

## Referee Comment (RC2) · Anonymous Referee #2 · 20 Apr 2020

In "Understanding the uncertainty in global forest carbon turnover" Pugh et al use remote-sensing based turnover estimates to evaluate the performance of six TBM. Based on this evaluation the authors propose eight hypothesis which are then discussed.

The study is well structured, the discussion is insightful and the hypotheses are supported by the analyses. It is clear that a lot of thinking went into this analysis which by itself is a sufficient reason to support publication of this manuscript.

In my opinion the discussion lacks one section, i.e., a critical assessment of the concept of biomass turnover and whether it is key benchmark for model evaluation or an observation that should only be used if more process-specific observations become available. Given that several model groups are replacing their turnover parameter by

an explicit representation of the different mortality events, what is the future of these remote-sensing based turnover estimates?

From a scientific point of view the manuscript could be accepted as it is. Nevertheless, the current manuscript is very dense. The manuscript could become easier to read and digest (and would therefore become more likely to make an impact) by: (1) Rewriting/expanding the equations (especially eq 2). The study does a good job in disentangling the major processes that contribute to the turnover time of biomass carbon but the equations fall short of reflecting this complexity. Either the introduction or section 2.1 could be used to refine and better formalize the definition of turnover. Ideally each of the hypothesis should be reflected in one of the terms shown in the final equation. (2) Rethink fig 1. I don't get the meaning/purpose of figure 1. I think it is related to my point above, i.e., showing the diversity of processes contained in the remote-sensing based turnover observations but it did not help me. Turning this figure into a table may help. After reading the entire manuscript, I think I would have benefited more from a description of each of the terms with an example rather than the bars and arrows. (3) Thinning the results section. In my opinion the model comparison is the least developed part of the manuscript and I even doubt whether it is essential. If the definition gets better developed, it might be possible to derive the hypothesis from the definition and then discuss these hypothesis in the light of scientific literature. This would change the type of study but it could increase the impact of this study. If you decide to keep the model comparison, please, better justify the model experiment (and add revision numbers for each of the models). It would have been much easier to compare the models if a run with a prescribed PFT distribution was used as well. How can you justify the comparison of data with management to simulations without management? How meaningful is this given that management is a major driver of both the growth and the mortality components of turnover? Given the complexity of the processes described by turnover but the simplicity of the observations (i.e. a single number), the model comparison remains superficial in the sense that it is hardly possible to label some of the model behavior as "very unlikely". In the end this section takes up a lot of space

for very little information (although I liked Fig 2 a lot. It is an informative way to show both models and data – note that this is the only figure that shows the observations). Maybe the bulk of the comparison could be moved to the supplementary materials?
* * *

---

## Author Comment (AC1) · 29 May 2020

We would like to thank the reviewer for their very careful reading of our manuscript which we are confident will lead to an improved paper.

This paper analyzes turnover of forest carbon simulated by six TBMs (terrestrial biophere models) using a common simulation protocol "to assess their current capability to accurately estimate biomass carbon turnover times in forests and how these times are anticipated to change in the future." (abstract). The concentration on turnover is motivated by referring to studies that according to the authors have shown that turnover time has "comparable or even larger importance than NPP when assessing the response of $C_{veg}$ to environmental change" (lines 69/70). The analysis proceeds by first setting up a list of potential mechanisms that might modify turnover time upon environmntal change (table 5). Then, splitting the turnover fluxes obtained in the different simulations into components arising from different mechanisms (phenology, mortality), the analysis of the simulation data is condensed into a set of hypotheses on the controls for the behaviour of turnover time seen in the different models. The main result of this analysis is that in the different models the dominant mechanisms controlling turnover are different so that more research is needed. To direct such research the authors discuss for their hypotheses on the diverse controls how the scientific community could proceed by making use of existing data and what the challenges are in achieving progress.

Overall, I really don't know what to make out of this paper. The result from the analysis of the different turnover fluxes that in the different models the overall turnover is caused differently is not much surprising since this diversity is "strongly linked to model assumptions concerning plant functional type distributions and their controls" (lines 47/48), which are different across models.

>> Indeed differences in simulated turnover fluxes among models are related to model assumptions. Here we go to some length to identify both how the model assumptions differ and the implications of these differences for estimates of turnover fluxes.

Making at all something out of this mess by coming up with a list of hypotheses on the different controls in the various models (summarized in the paper title somewhat overstated as "understanding the uncertainty"), is a rather heroic act, demonstrating the author's solid knowledge on ecological processes and their implementation into models. But then selling this mess by coming up with recommendations for the community on how to proceed makes the whole paper resembling an (extended) introduction for an application for funding rather than being a text informing the interested comunity on advances in understanding. I do not deny that this type of reasearch must be done to advance – and I appreciate the author's sophistication in their analysis – but it seems to me research that needs to be done before performing the true research because it is "only" formulating questions and setting up the research program. Accordingly, I am not convinced that such type of research that stops where it starts to get interesting is of sufficient interest for a larger community to be published in a reviewed journal instead of simply staying a paper in the grey literature.

>> TBMs are complex and widely-applied models, whose results underlie major scientific and policy-relevant efforts, such as assessing the current and future strength of the terrestrial carbon sink (Ciais et al., 2013; Friedlingstein et al., 2019; Huntingford et al., 2013; Jones et al., 2013). Virtually every study raises new questions for further research and in some cases the effort required to identify those questions can be substantial. The aim here is to go beyond an intercomparison that simply reports differences, to identifying why those differences occur, including comparisons with observational data. Understanding the nature and origin of the differences is important to further provide future robust projections beyond the "black box" of carbon/climate models. The discussion is intended as a perspective on how some of the uncertainty we identify could be reduced in future work.

We therefore contend that identifying why the models differ so widely in their turnover responses and what can be done to reduce these uncertainties is an important and non-trivial scientific effort. We hope that our study is indeed useful to the scientific community, particularly the wide community that develops and applies TBMs and makes use of their outputs.

Otherwise the paper is well readable, although the structure of the paper, outlined in the first paragraph above, could be emphasized stronger within the main text of the paper to keep the reader aware of the purpose of the different sections within the overall construction of the paper so that the 'red line' is better visible; e.g. only later I realized that the aim of section 3 (Results) is to identify the hypotheses listed in table 5 that subsequently, in the discussion part (section 4), are further discussed.

>> We agree with this good suggestion and have added the following text at the end of the introduction to make the structure clear.

*"We first analyse historical vegetation carbon turnover time estimates from the models, comparing the models with a limited selection of available large-scale constraints and identifying implicit or explicit model-based hypotheses that may explain why the estimates diverge (§3.1). We then identify hypotheses behind differing future turnover time estimates based on an exemplary strong climate change scenario (§3.2). Finally, we present a discussion as to how these hypotheses can be tested to advance understanding of turnover times, building on available data sources where possible (§4)."*

One other general issue concerns the specification of author contributions (Lines 605- 7): Looking through the list, 3-4 authors remain that only commented on the manuscript without indication of any significant contribution to it. According to good scientific practice, this is not sufficient to justify authorship (otherwise I as reviewer should also be listed as author): The European Code of Conduct for Research Integrity [1] specifies "...authorship ...is based on a significant contribution to the design of the research, relevant data collection, or the analysis or interpretation of the results.". Hence for this paper authorship should be reconsidered. (And btw: who is "KZ"?)

>> Thank you for catching that we have left some authors out of the contribution section. We have remedied this. It now reads.

*"TAMP conceived the analysis. TAMP, JB, BB, VH, AH, JH, KN, AR and HS contributed model simulations. TAMP, TR, SLS, and JS carried out the data analysis and all authors contributed to data interpretation. TAMP, AA, SH, MK, BQ, TR, SLS, BS, KT and TH wrote the manuscript. All authors commented on the manuscript."*

KZ should have read KN

(1) Underlying this study is the general picture that vegetation carbon is growing due to NPP and diminishing due to "turnover" fluxes called collectively $F_{turn}$. This is embodied in the first part of eq. 1, the rate equation $dC/dt = NPP - F_{turn}$. During the analysis presented in the paper, $F_{turn}$ is further subdivided into fluxes arising from different processes. Hence underlying this analysis is also a formula like $F_{turn} = F_1 + F_2 + F_3 + \ldots$ Moreover, for each such partial turnover flux a separate characteristic time is defined by a formula like $\tau_i := C_{veg}/F_i$. All this structure is not made explicit and transpires only while reading. I think for a better readability it would be helpful to make this structure explicit right at the beginning, maybe directly in connection with eq. 1. This would also be the right place to explain in detail the processes underlying all the partial turnover fluxes distinguished in this study, because these explanations are scattered across the paper (what is eg. "Background" in Fig. 6?).

>> We agree with this point, which was also raised by Reviewer 2. We have added additional equations to the introduction:

*"Turnover time of existing biomass can thus be calculated as,*

*$\tau = C_{veg}/F_{turn}$                (Eq. 2),*

*$F_{turn}$ is the total loss flux of live biomass due to the transfer of plant tissue to dead pools of litter and soil, to harvest products and residues, or to the atmosphere via burning. It can be decomposed into,*

*$F_{turn} = F_{mort} + F_{leaf} + F_{fineroot} + F_{repro}$            (Eq. 3),*

*where $F_{mort}$ is the carbon turnover flux due to plant mortality or woody carbon loss, $F_{leaf}$ and $F_{fineroot}$ that due to leaf and fine root senescence respectively, and $F_{repro}$ turnover due to reproductive processes (e.g. flowers, fruits)."*

We have also expanded the description in section 2.1:

*"Turnover times can also be defined relative to particular turnover fluxes, such as those outlined in Eq. 3. In this case the turnover time is calculated respective to the appropriate biomass pool, i.e. turnover time of vegetation biomass due to mortality, $\tau_{mort}$, is defined as $C_{veg}/F_{mort}$, and turnover time of fine root biomass, $\tau_{fineroot}$, is defined as $C_{fineroot}/F_{fineroot}$, where $C_{fineroot}$ is the fine root biomass. $F_{mort}$ can also be decomposed into fluxes resulting from particular mortality processes, for instance, following the conceptual groupings in Table 3,*

*$F_{mort} = F_{mort,vitality} + F_{mort,disturbance} + F_{mort,background} + F_{mort,heat} + F_{mort,other}$            (Eq. 4),*

*although other process breakdowns can also be applied. Accordingly, a turnover time can also be defined for $C_{veg}$ relative to each mortality process, e.g. $\tau_{mort,vitality} = C_{veg}/F_{mort,vitality}$. Turnover rates are the inverse of turnover time, i.e. $1/\tau$."*

We do not think it practical to represent the hypotheses directly in these turnover equations. The hypotheses relate to how individual turnover fluxes are simulated in the TBMs, each of which has very different formulations (e.g. Table 3). We have modified the caption of Figure 6 to reference Table 3 (it was previously erroneously linked to Table 2) and to explicitly list all the 5 groupings.

(2) Furthermore, for the analysis framework it is important to note that all partial timescales are defined with reference to the same carbon mass $C_{veg}$. By this common reference the flux decomposition can also be written as $F_{turn} = C_{veg} (\tau_1^{-1} + \tau_2^{-1} + \tau_2^{-1} + ...)$ so that the rates $1/\tau_i$ indicate the size of the relative contributions of the different turnover processes to the total turnover flux. Since it is hardly believable that for all the different turnover processes the fluxes are proportional to the total biomass $C_{veg}$, the meaning of those time scales $\tau_i$ is only diagnostic and may have nothing to do with the actual time scales underlying the various processes causing turnover. An example is fires that depend on the amount of fuel (litter or crown biomass) but not on stem biomass that contains the majority of vegetation biomass. This poses a serious question mark on the overall approach taken in this study by putting time scales into the center of the analysis. – Nevertheless, ratios of the diagnostic time scales stay well interpretable as ratios between turnover fluxes because in this framework $\tau_i^{-1}/\tau_i^{-1}$ = $F_{turn}/F_{turn}$.

>> We did not previously clarify how the partial timescales are calculated (please see response to point above). We have corrected this oversight and included additional text for clarification. However,

the general concern raised here is one of scaling, i.e. at what level is it appropriate to conceptualise the turnover time. We agree that there is more than one possible solution. We employ the common definition of turnover time at the whole plant scale, i.e. relating to $C_{veg}$, for $\tau_{turn}$, $\tau_{NPP}$ and $\tau_{mort}$. As the reviewer states, these summarise over several sub-processes, but provide a useful summary of the net effect of these sub-processes and are readily comparable to other studies in the literature. In diagnosing the causes behind whole plant turnover time we calculate turnover times or rates for the sub-processes based on the mass of the component upon which they act. For instance, for fine roots, we calculate relative to the biomass of fine roots. Please note that we report in all cases turnover fluxes, and thus turnover times, of live vegetation. For instance, in the case of fires, we report mortality due to fires, not the carbon combusted by fires. This is consistent with the other mortality processes for which we report the carbon in trees that die, not the timescale over which it decomposes.

(3) For the analysis done in this paper the authors hop between comparing fluxes $F_i$ (e.g. Figs. 6 and 7d) characteristic times $\tau_i$ (Table 4, Fig. 7), rates $\tau_i^{-1}$ (Figs. S11-S15), and the additional time scale $\tau_{NPP}$ := $C_{veg}$/NPP (Fig. 1). It would be good to have a discussion on the advantages of using the one or other quantity. Anyway it should be made transparent why $F_i$, $\tau_i$, $\tau^{-1}$, or $\tau_{NPP}$ are chosen in the different parts of the analysis and why not the respective others.

>> Generally, we consider turnover times to be the most intuitive metric to interpret the results herein. For certain components of the analysis (e.g. Figs. 4, 6, 7d) it is more appropriate to work with the fluxes themselves as we are addressing how the fluxes used to calculate turnover time are constituted. We present rates for the decomposition by mortality process in Figs. S11-16 because we find the turnover time concept to become increasingly abstract at these partial breakdowns. In the revised Section 2.1 we have made explicit how each of these metrics relate to each other. We have only included $\tau_{NPP}$ for comparison with satellite-derived estimates - this is explained in Section 2.1.

(4) The descriptions in the main text heavily draw on plots in the supplement and there seems to be no systematics in putting plots here or there. E.g. for all the many claims on p. 11 the reader is referred to plots in the supplement which is in my opinion a good reason to move those plots – maybe in aggregated form – into the main text. And if the authors stick to their decision to have Fig. S9 in the supplement, then also the description on how to arrive at that figure (lines 221-229) should be shifted to the supplement. In conclusion, it seems to me that the readability of the paper could be improved by reconsidering the division of plots between main text and supplement.

>> Thank you for this point. We have tried to include figures in the main text that underline important results relating directly to the derivation of the hypotheses. We agree that Figure S9 would fit well in the main text. We also reflect that Figure 8 is not crucial to the main points in the text and can be moved to the supplement. We agree that page 11 references Figs. S11-S16 several times and that it would be helpful to have a summary in the main text. We have chosen needleleaf evergreen and tropical broadleaf forest types as exemplars and included plots for these forest types in the main text in additional to the supplementary plots.

(5) As already mentioned above, the study is motivated by the claim that $\tau$ has "comparable or even larger importance than NPP when assessing the response of $C_{veg}$ to environmental change" (Lines 69-73). Because this is the major justification of this study, the authors should be more precise here. E.g. Ahlström et al. (2015a) demonstrate that for stationary conditions turnover has only about 2/3 the importance of NPP (l.c. Fig. 3c), and should (to my understanding) be even less important for the non-stationary conditions of environmental change one is actually interested in. Even better would be to add further plots to this study demonstrating explicitly the author's claim by separating contributions to the biomass changes from NPP and turnover; an analysis in the spirit done by Friend et al. (2014)

applied to the transient simulations in the paper would do it. To add: Simple reference to that Friend et al. paper will not be sufficient because their main message is so cryptically formulated that it is inconceivable (they write: "The variance in final $C_{veg}$ caused by differences in fitted residence time relationships between models was found to be 30% higher than that caused by differences in the fitted NPP responses when all models were considered." (p. 3283). What 'variance' is meant here, and why only "when all models are considered"?)

>> We seek here to understand the reasons behind different estimates of turnover time, not attribute its relative importance compared to other drivers of $C_{veg}$. The attribution of Friend et al. (2014) clearly demonstrates the greater influence of differences in turnover time than NPP on modelled $C_{veg}$ in 2100. It is not our place to clarify their meaning in the quote above, but the following sentence from their paper is quite unambiguous, "*In other words, for the non-HYBRID4 models, differences in residence time relationships with climate between the models are responsible for more than twice the variation in modeled global $C_{veg}$ change to 2100 than are differences in NPP relationships with temperature and $CO_2$.*"

Ahlström et al. (2015) found that variations in $\tau$ were a major driver in the variations in $C_{veg}$ caused by different environmental forcing. Thurner et al. (2017) did not make a formal decomposition of the relative importance of NPP and turnover time but found modelled NPP to usually be close to the observation-based estimates, whereas biomass usually was not, indicating the role of turnover time. Galbraith et al. (2013) show a very wide variation in estimates of woody biomass turnover time by TBMs for the tropics. Johnson et al. (2016) show a clear relationship of $C_{veg}$ with stem mortality rates, but not with NPP. We thus stand by our statement as written.

Friend et al. (2014) already demonstrated for a broadly comparable set of TBMs that differences in turnover time had more influence on $C_{veg}$ than did NPP and we see little to gain from repeating that analysis here.

"Parazoo et al. (2018)" was a miscitation and has been removed.

Line 64: Here you introduce as key term the word 'residence time', nowhere else used in the paper, and the term 'turnover time' used throughout the paper, is only introduced in brackets. It's better to drop 'residence time' because its meaning is anyway ambiguous (see Sierra et al. 2017).

>> We have dropped the term as suggested.

Lines 75/76: For stationary states I agree that there are only these two causes, but for nonstationary conditions another one appears: Once NPP is changing, the internal memory causes a mismatch between NPP and $F_{turn}$ so that the turnover time $\tau_{turn}$, which is only a diagnostic obtained from $C_{veg}/F_{turn}$, changes in addition to the causes mentioned because the input NPP changes. I think the authors are aware of this (see lines 141/42 where the difference between $\tau_{NPP}$ and $\tau_{turn}$ is addressed), but since one is ultimately interested in the nonstationary case of environmental change this should not be overlooked.

>> Whilst changing NPP affects $\tau_{NPP}$ because the flux into the vegetation is no longer in balance with $C_{veg}$, $\tau_{turn}$ is not affected because it simply defines the rate of loss of $C_{veg}$. To demonstrate this, we define a simple single-pool model of $C_{veg}$ in which NPP is the flux into $C_{veg}$ and $F_{turn}=C_{veg}/\tau_{actual}$ defines the flux out of the pool, i.e. there is a constant rate of loss defined by $\tau_{actual}$. We increment NPP for 500 years and then hold it constant thereafter (Fig. R1a). Actual turnover rate is held constant, i.e. $\tau_{actual}$=50 years throughout. $F_{turn}$ lags NPP during the period in which NPP is incremented due to the memory effect. $\tau_{NPP}$ shows a moderate deviation from $\tau_{actual}$ (Fig. R1c) as NPP and $F_{turn}$ become unbalanced.

However, $\tau_{turn}=\tau_{actual}$ throughout. Unless the change in NPP directly influences the turnover rate, $\tau_{turn}$ is not affected by changes in NPP. This latter is a special case which does occur in some of the TBMs here, but is broadly captured by the term, "response to environmental variation". We prefer not to complicate this paragraph with a caveat on this point, but retain the text in Section 2.1 noting the difference between $\tau_{turn}$ and $\tau_{NPP}$.

[Figure]

Figure R1. Results from a simple single-pool model of $C_{veg}$ demonstrating the effect of changes in NPP on $\tau_{NPP}$ and $\tau_{turn}$.

Throughout the paper the term "vitality" is used 28 times to refer to a certain group of processes. In lines 95/96 two of them are named, but there is no explanation on the general notion of this term. But such a general notion is needed to understand e.g. the claims in lines 97, 119, 295-298 etc. To me a formulation like "vitality-related mortality" (lines 320/21) sounds a bit weird: Increased mortality improves vitality? Without further help I have problems to make sense out of this vitality concept.

>> We have modified the text previously starting on line 95 to read as follows:
*"Mortality is also commonly prescribed to occur linked to a negative carbon balance or if tree vigour is low (for instance, if growth efficiency, the ratio of NPP to leaf area, falls below a defined threshold). In principle, such formulations should capture both environmental stress and competition with neighbours, but in some TBMs such processes are supplemented or replaced by self-thinning rules to represent the effect of competition (e.g. Haverd et al., 2014; Sitch et al., 2003). Herein, all mechanisms related to carbon balance, vigour or competition are broadly termed "vitality-based", in that they directly or indirectly relate to the vital status of the tree."*

I find it hard to memorize the meaning of your notation 'MI', 'MP', 'MS'. Maybe you indicate why you choose these combimations of letters or switch to a more mnemonic notation.

>> These are intended to refer to mechanisms acting at the individual level (MI), stand level (MS) and population level (MP), with the M indicating "Mechanism" (to avoid confusion with other notation) and the second letter indicating individual, stand or population. We have changed the sub-headings in Table 1 to try and make this more intuitive, e.g. "Mechanisms at individual-level (PFT) (i.e. functional composition unchanged) (MI)"

Table 1: It took me really a hard time to understand how columns 2-4 should be read, the caption is in this respect rather unclear. Now I got that these are a pictorial representation of the assumptions on the topics in the headline. To prevent that other readers also struggle with this, I suggest to have another headline covering columns 2-4 eg. with title "assumptions on" and columns 5 and 6 with title "consequences for". Nevertheless, I still do not understand how to read column 2 "Resource capture (NPP) and allocation to woody or soft tissue": Has the combined length of the bars for "Soft" and "Wood" any meaning in relation to some maximum length, e.g. the largest combined bar or the width of the column? Or do you want to emphasize the relative lenghts of the bars "Soft" and "Wood"? Does a "Wood" bar shorter than the associated "Soft" bar mean that the influence on the former is smaller?

>> We apologise for not making this clearer. We have implemented the nice idea to clarify the meanings of columns 2-4 and 5-6 with additional headlines. We have added an explanation of the mean of the length and shading of the bars to the column heading, "Level of NPP (bar length) and allocation to woody or soft tissue (bar shading)".

Table 1: In some cases the consequences for biomass and τ shown in columns 5 and 6 seem to me rather speculative and do not follow directly from the information given in columns 1-4. Therfore this table should be revised or at least discussed in detail somewhere. My concerns on the proposed mechanisms are:

>> In the below, it appears that much of the confusion relates to the examples given. We wish to emphasize that this figure is intended to breakdown the different mechanisms by which biomass and τ can change. In reality or in models, several of these processes may change at once for a given perturbation. As such, the examples are only illustrative and not intended to suggest that the only effect of that driver is through the mechanism it is listed against. This has been clarified in the caption. We have also added the following text to the introduction where Table 1 is introduced:

*"The individual mechanisms within these groupings are isolated within Table 1 so as to show how a particular perturbation in NPP, allocation, or turnover rate of woody or soft tissues (e.g. leaves, fine roots and fruits) would affect biomass or τ. Because both trees and ecosystems respond to environmental stimuli in a coordinated fashion, it is likely that many of these mechanisms will occur in concert."*

$MI_{NPP,F}$: Upon rising NPP it is claimed that τ remains unchanged. This is only true for very slow changes in NPP so that the system essentially stays in equilibrium. For a faster increase of NPP the turnover flux stays behind NPP because of the internal memory so that $\tau_{turn} := C_{veg}/F_{turn}$ changes even though only the input to the system changes. (See also comment on lines 75/76 above.)

>> We disagree here. Although $F_{turn}$ does lag NPP, the change in $C_{veg}$ also lags NPP, offsetting the lag in $F_{turn}$. Please see our response to the previous comment, above.

$MI_{RA}$: A typical experience when experimenting with water stress in TBMs is that by increasing water stress, in dry regions annual production may increase. The reason is that the resulting reduced NPP flux leads to water savings so that the growth period is lengthened, to the consequence that the annual production (accumulated NPP) rises. For your table entry this would mean that you can as well reverse the direction of your biomass arrow. What the consequences for τ are would be hard to say.

>> Our intention here is to show the effect of a change in the relative fraction of resources allocated to wood vs soft tissues, not the wider effects of a water stress event per se. We have modified the example column to be more specific.

$MI_{ST}$: The same remark on the counter-intuitive reaction on increased water stress applies also here so that biomass may increase. On the other hand: By increasing exudates it would also be plausible that biomass would decrease.

>> As above, we attempt to give an example of what might cause a change isolated in the left four columns. Here we agree, however, that exudation is not the clearest example, which we have instead changed to increased levels of herbivory.

$MP_{MR}$: For the example driver "Reduced defensive investment" the consequences could be indeed as you describe if the investment was optimal or sub-optimal before. But in case of super-optimal

investment in defense, biomass could as well rise so that also the converse behaviour of biomass and turnover rate is conceivable. And, just to note, it doesn't matter whether the reduced investment in defense arises from a shift in functional composition or not.

>> As above, this is just intended to be one example of why a different species mix might have higher turnover rates from mortality.

$MP_{NPP}$: I do not understand what is meant by "intrisic NPP" and in what respect the example driver is assumed to be "conservative".

>> We meant a change in NPP due to a change in species. On reflection, we agree "intrinsic" is confusing and have removed it. We have changed the example to make it clearer, "New species mix has different respiratory costs, e.g. for defense". We also make this change to further indicate that a particular driver may act on several mechanisms.

$MP_{RA}$: I do not see why a shift to species with reduced wood density may necessarily decrease the characteristic turnover time: This is – as far as I see – only true when those species with reduced wood density have themselves lower than average turnover time. If not, turnover time may as well increase.

>> Reduced wood density could allow the same height and volume of tree for lower construction cost. We agree that the well-established association of wood density with mortality risk makes this example confusing. We have replaced it with, *"Lower specific leaf area in new functional mixture".*

$MP_{ST}$: In the table you claim that upon an increased phenological turnover rate due to a shift in functional composition biomass stays unchanged. This seems to me rather unintuitive: This assumes that all species have the same productivity per leaf or root area ("effectivity"), which is typically not the case. Therefore, if the composition is shifted to more effective species the overall LAI or root area may decrease but the total productivity and therefore biomass may increase. And also the other direction is conceivable.

>> We are claiming that a shift in phenological carbon turnover rate alone would not substantially affect biomass (there would be a small shift in leaf biomass; we explicitly exclude this in the revised version by referring to woody biomass). Other changes associated with a change in phenological turnover may affect biomass, e.g. a shift towards a deciduous phenology may increase phenological carbon turnover, but be associated with a change in allocation ($MP_{RA}$) towards soft tissues to fund this, or with a reduction in LAI leading to a lower NPP ($MP_{NPP}$).

Lines 186-189: It is unclear why this logic for identifying the forest mask was chosen. What goes wrong when one of the conditions (e.g. the condition on boreal PFT) would be omitted?

>> The logic comes from Hickler et al. (2006) and Smith et al. (2014). These studies were referenced here, but we have adjusted the phrasing to make it clearer that we follow their logic.

Line 235: Wilkoxan → Wilkoxon.

>> Corrected.

All figures: Put units at color scale.

>> We have added units to the figures themselves as requested.

Figure 2: According to the caption the figure shows "density kernels". In the caption you explain "density", but the term "density kernel" is nowhere explained. I guess that you show the global relative abundance of $\tau_{NPP}$ values. Please consider a renaming or explain the term "density kernel" somewhere.

>> "Density kernel" is synonymous with probability density distribution here. We changed the references in the text and caption to use this more common expression.

Line 260: Here you claim that your findings are summarized in the alternative hypotheses H1a and H1b listed in table 5. But in the table entry you link the allocation fraction of NPP to soft tissues or wood to $\tau_{mort}$, but in the text leading to your claim you argue with the turnover fluxes and neither NPP nor $\tau_{mort}$ are mentioned, so that the link between text and hypotheses is missing. Moreover, I do not understand why there should be such a close link between the allocation fraction and $\tau_{mort}$ as you claim in either of the two variants of the hypotheses H1 in the table text, it is easily imaginable that allocation fraction and $\tau_{mort}$ are completely unlinked.

>> We have substantially revised the text here to make the link explicit. We had placed the sentence equating $F_{turn}$ to NPP at the start of the next paragraph, but have now brought it forward to this paragraph to improve the logical flow of the text.

*"Consistent with the logic that $F_{turn} \approx NPP$ (Section 2.1), the partitioning of $F_{turn}$ among tissue types is approximately equal to the allocation of NPP between those tissue types. For no change in overall structure, a fraction of $F_{turn}$ resulting from leaf, fine root or reproductive turnover implies the same fraction of NPP must be invested in the corresponding tissues. Therefore, to maintain a given biomass for a given NPP, the results in Fig. 3 reflect two distinct hypotheses linking allocation of NPP to $\tau_{mort}$. Either a large fraction of NPP is invested into wood, resulting in $F_{mort}$ being a large fraction of $F_{turn}$ and thus implying a relatively low $\tau_{mort}$, or a relatively low fraction of NPP is invested into wood, resulting in $F_{mort}$ being a relatively small fraction of $F_{turn}$ and thus requiring a higher $\tau_{mort}$ in order to maintain the same biomass (Table 5: H1a and H1b)."*

Lines 260/61, 314/15 and Fig. 4: At those lines you make clonclusions from Fig. 4 on the relative contributions of $F_{phen}$ and $F_{mort}$ to the variability of $F_{turn}$. I doubt that such conclusions can indeed be drawn from that figue. Such conclusions could be drawn if the fluxes $F_{phen}$ and $F_{mort}$ would be uncorrelated, but because $F_{phen} + F_{mort} = F_{turn}$ a large phenological flux leaves not much room for variation in the mortality flux and conversely. Therefore those two fluxes should be strongly (inversely) correlated and the condition of vanishing correlation to draw your conclusions is not met. Accordingly, I think this analysis should be discarded.

>> There is no limit on the sum of $F_{mort}$ or $F_{phen}$, as such a high value of one does not imply a lower value of the other. The two fluxes are positively correlated in all models here. However, in order to tie in more closely with H2, we have replaced Fig. 4 with a new calculation that explicitly assesses the effect of the local deviation in $F_{mort}$ on the local value of $\tau$ and then summarises these effects on $\tau$ across all grid cells (Fig. R2). The results lead to exactly the same conclusions as our previous Fig. 4a.

The new method is described as follows:
***Contribution of turnover fluxes to spatial variation in $\tau$:*** *Following Eqs. 2 and 3, $\tau_{turn} = C_{veg}/(F_{mort} + F_{phen})$, where, $F_{phen} = F_{leaf} + F_{fineroot} + F_{repro}$. $\tau_{turn}$ was calculated for each grid cell with at least 10% forest cover. $\tau_{turn,fixmort}$ was then calculated in the same way except for replacing the local value of $F_{mort}$ with its mean across all grid cells. The difference between $\tau_{turn}$ and $\tau_{turn,fixmort}$ provides the difference in $\tau_{turn}$ due to the local deviation in $F_{mort}$. The results were summarised at global level by taking the mean*

*absolute deviation of $\tau_{turn}$ - $\tau_{turn,fixmort}$ across all grid cells. The same procedure was carried out to assess deviation due to $F_{phen}$.*

[Figure]

*Figure R2. Mean absolute deviation in $\tau_{turn}$ across all grid cells with at least 10% forest cover as a result of using global mean values of mortality ($F_{mort}$) or phenology ($F_{phen} = F_{leaf} + F_{fineroot} + F_{repro}$) turnover fluxes in the calculation of $F_{turn}$ in Eq. 2 (see Methods). Larger values indicate a greater contribution of $F_{mort}$ (blue) or $F_{phen}$ (green) to spatial variability in $\tau_{turn}$. Calculated over the period 1985-2014 from the CRU-NCEP-forced simulation.*

We have removed Fig. 4b after reflecting that the relevant point in the text is made more effectively using Fig. 7d.

Line 265: Where did you already follow "the same logic"?

>> This logic was explained in Section 2.1 where the definition of $\tau_{NPP}$ was given. We have added a reference to this section to the statement.

Line 295: What is the reason to start the sentence with "However"?

>> Because the previous sentence refers to a wide range of approaches used to represent mortality in models, whilst this sentence indicates that despite this, there are similarities in the general categories of processes. We have changed the beginning of the sentence to, *"Despite this diversity, there are similarities in the broad categories of processes included. All models..."*

Fig. 6: What is "Background"? In the caption: "vertical axis" → "vertical side bar"

>> "Background" has been clarified in the caption, *"'Background' covers mortality based on a fixed rate or tree age."* A reference to Table 3 has also been included. We have changed the labelling for the side bar as suggested.

Fig. 7: I am not sure what you really show in subfigure d: In the caption you write that you show the "Fraction of total turnover due to mortality" but all curves start at zero so that some change is shown. Maybe you show the relative change in $F_{mort}/F_{turn} = \tau_{turn}/\tau_{mort}$? If so it would be more clear to write down this formula. I also don't know how to understand the in-figure text "Δ Fraction as mort." – why "as"?

>> We have added the formula to the caption as suggested and relabelled panel d as suggested. We have also added the clarification, *"All plots show relative changes compared to a 1985-2014 baseline."*

When distinguishing between PFTs it remains unclear whether characteristic times (Figs. 2, 5) and rates (Figs. S11-S15) are calculated with reference to total $C_{veg}$ obtained for the whole mix of PFTs in

a grid cell, or with reference to the $C_{veg}$ of the individual PFTs in a grid cell. This difference has e.g. a large impact on whether changes displayed in Fig. 8 can be interpreted on the basis of Figs. S11-S15, as done on p. 11.

>> For Figure 2 and 5 we applied the forest type classification to determine the dominant PFT, used the entire $C_{veg}$ for the grid cell, but weighted NPP by the fraction of forest cover in each grid cell (as described in Section 2.4). A core purpose of Figure 2 is to compare TBMs to the remotely-sensed data, which does only give us a value for the entire grid cell $C_{veg}$, so using the entire grid cell $C_{veg}$ is most consistent for the purpose of comparison. The purpose of Figure 5 is to compare total mortality derived turnover times across TBMs for grid cells classified as one specific forest type, thus is makes sense to use the total biomass and mortality flux for each grid cell. To clarify this, we have added the following sentences to the two captions, respectively:
*"For the models, $\tau_{NPP}$ was derived from entire grid cell $C_{veg}$ and forested area-weighted NPP, as for the satellite-based product (see Section 2.4). "*
*"$\tau_{mort}$ was derived from entire grid cell $C_{veg}$ and forested area-weighted NPP, as for the satellite-based product (see Section 2.4) and grid cell were classified according to dominant PFT."*

Figs. S11-S15 are based on total $C_{veg}$ for the whole mixture of PFTs in the grid cell. Please note that all of these figures are based on forest type maps, please see the description of forest type classification in Section 2.4. We have added clarifications to the caption of each figure.

Lines 339-41 and hypothesis H5/H6 in table 5: To me the description in lines 339-41 is inconsistent with the resulting formulation of hypotheses H5 and H6. In line 340 it is said that LPJmL's "increased mortality of established trees" is the reason for the shift in PFT composition – hence LPJmL cannot fall under H5, where "changes in turnover rate of individual PFTs" is explicitly excluded as cause, and the latter exclusion covers the former. Since LPJ-GUESS is listed to fall under H5 and H6 (which I do not understand) I guess that there is some insufficient distinctness in the formulations of the text and the hypotheses that make it impossible to follow what is meant here.
Lines 347/48: Here it is claimed that JULES "implicitly" falls under hypothesis H5. But why is it then not listed in table 5 under H5?

>> We were attempting to capture a subtlety between changes in turnover time in the long term resulting from compositional change (which also apply to LPJmL) versus sustained reductions in turnover time of an existing PFT. However, we accept that this was not clear and is in fact hard to define with the results here. The subtlety we were originally aiming to show is now mentioned in Section 4.5 of the discussion. We have reformulated H5, splitting it into two sub-hypotheses and subsuming H6 within it.

>> H5a: Environmental change leads to large changes in the mortality rates of associated with PFTs, which dominate the change in $\tau$ over the 21$^{st}$ century.

>> H5b: Shifts in forest functional composition, rather than changes in the turnover rate associated with PFTs, dominate the response of $\tau$ to environmental change over the 21$^{st}$ century.

LPJmL is assigned to H5a, LPJ-GUESS and JULES to H5b.

Lines 383-386: I guess "(1)" refers to MI and MS in table 1, while "(2)" refers to MP in that table. Its a bit irritating that you claim to have "identified" (line 383) these two groups "(1)" and "(2)" – in table 1 it were three groups and they were not "identified" but postulated. How did you "identify" these two goups?

>> As the reviewer rightly points out, these responses were postulated in Table 1. They are then identified throughout the results in Section 3.2, as indicated in the numerous references to these mechanisms in that section. We have made some small changes to the wording in the text referred to here:

*"As postulated in Table 1, two contrasting modes of simulated turnover response to changing environmental conditions were identified in the simulations: (1) individual or stand-level responses where internal physiology influences turnover in response to temperature, atmospheric $CO_2$ concentration, or other extrinsic drivers (MI, MS mechanisms); and (2) population responses where shifts in species composition, age distribution, etc. influenced forest composition or demography, with concomitant changes to turnover (MP mechanisms)."*

Lines 389-391: I have a problem to understand this sentence: What is meant by "These differences"? Differences because $MI_{ST}$ and $MP_{NPP}$ don't show up? "These" cannot refer to differences in the process diversity implemented into the models, because such differences were not addressed before. Hence you mean differences between implemented processes and "emergent response"? This makes no sense. I am lost.

>> This sentence was not well formulated. We have changed it to:
*"The diversity in both the processes that are included in models and the simulated emergent response in turnover time, arise largely because the key ecosystem states and fluxes, and their relationships to environmental drivers, are under-constrained by observations at regional and global scales."*

And what means "have been under-constrained"? So they are constrained now?

>> There is now potential to constrain many of them, as we outline in the discussion. Nonetheless, we have changed this to, *"are under-constrained".*

Lines 403-405: Wrong grammar: ". . . however . . . which is usually absent . . . "?

>> Corrected.

Line 458: "that is key" → "is key"?

>> We have removed the first "that".

Lines 563/64: The construction of the sentence is a bit weird. Maybe better:
". . . conditions (H1, Section 4.1), it is perhaps not surprising that the TBMs show different responses of allocation to increased productivity following mechanisms $MI_{NPP,F}$ or $MI_{NPP,FS}$.".

>> Changed as suggested

**References**

[revised manuscript text omitted]

---

## Author Comment (AC2) · 29 May 2020

We would like to thank the reviewer for their helpful suggestions which we are confident will lead to an improved paper.

In "Understanding the uncertainty in global forest carbon turnover" Pugh et al use remote-sensing based turnover estimates to evaluate the performance of six TBM. Based on this evaluation the authors propose eight hypothesis which are then discussed. The study is well structured, the discussion is insightful and the hypotheses are supported by the analyses. It is clear that a lot of thinking went into this analysis which by itself is a sufficient reason to support publication of this manuscript.

In my opinion the discussion lacks one section, i.e., a critical assessment of the concept of biomass turnover and whether it is key benchmark for model evaluation or an observation that should only be used if more process-specific observations become available. Given that several model groups are replacing their turnover parameter by an explicit representation of the different mortality events, what is the future of these remote-sensing based turnover estimates?

>> This is an interesting point. Biomass turnover is a metric that captures high-level dynamics of forest carbon cycling, but we agree that the processes that drive carbon turnover are much more nuanced. Accurately simulating the response of biomass turnover to different environmental conditions likely requires breaking down the underlying processes, such as we pick apart herein. However, any constraint on overall biomass turnover is also an important constraint on models. As the manuscript is already quite long, we prefer not to add a whole section on this, but we have added the following sentences to the start and end of the conclusion:

*"Biomass carbon turnover time is a high-level metric that integrates over a wide variety of underlying processes."*
*"This benchmarking must go beyond the emergent property of turnover time, to the underlying processes, facilitating model improvement as well as evaluation."*

From a scientific point of view the manuscript could be accepted as it is. Nevertheless, the current manuscript is very dense. The manuscript could become easier to read and digest (and would therefore become more likely to make an impact) by: (1) Rewriting/expanding the equations (especially eq 2). The study does a good job in disentangling the major processes that contribute to the turnover time of biomass carbon but the equations fall short of reflecting this complexity. Either the introduction or section 2.1 could be used to refine and better formalize the definition of turnover. Ideally each of the hypothesis should be reflected in one of the terms shown in the final equation.

>> We presume that the reviewer is referring to $\tau = C_{veg} / F_{turn}$ as given in the first line of section 2.1. Reviewer 1 also commented on this point and we agree that it would be clearer to expand the equations to split $F_{turn}$ up into its constituent components. We have added additional equations to the introduction:

*"Turnover time of existing biomass can thus be calculated as,*

$\tau = C_{veg}/F_{turn}$         *(Eq. 2),*

*$F_{turn}$ is the total loss flux of live biomass due to the transfer of plant tissue to dead pools of litter and soil, to harvest products and residues, or to the atmosphere via burning. It can be decomposed into,*

$F_{turn} = F_{mort} + F_{leaf} + F_{fineroot} + F_{repro}$         *(Eq. 3),*

*where $F_{mort}$ is the carbon turnover flux due to plant mortality or woody carbon loss, $F_{leaf}$ and $F_{fineroot}$ that due to leaf and fine root senescence respectively, and $F_{repro}$ turnover due to reproductive processes (e.g. flowers, fruits)."*

We have also expanded the description in section 2.1:

*"Turnover times can also be defined relative to particular turnover fluxes, such as those outlined in Eq. 3. In this case the turnover time is calculated respective to the appropriate biomass pool. I.e. Turnover time of vegetation biomass due to mortality, $\tau_{mort}$, is defined as $C_{veg}/F_{mort}$, and turnover time of fine root biomass, $\tau_{fineroot}$, is defined as $C_{fineroot}/F_{fineroot}$, where $C_{fineroot}$ is the fine root biomass. $F_{mort}$ can also be decomposed into fluxes resulting from particular mortality processes, for instance, following the conceptual groupings in Table 3,*

*$F_{mort} = F_{mort,vitality} + F_{mort,disturbance} + F_{mort,background} + F_{mort,heat} + F_{mort,other}$      (Eq. 4),*

*although other process breakdowns can also be applied. Accordingly, a turnover time can also be defined for $C_{veg}$ relative to each mortality process, e.g. $\tau_{mort,vitality} = C_{veg}/F_{mort,vitality}$. Turnover rates are the inverse of turnover time, i.e. $1/\tau$."*

We do not think it practical to represent the hypotheses directly in these turnover equations. The hypotheses relate to how individual turnover fluxes are simulated in the TBMs, each of which has very different formulations (e.g. Table 3).

(2) Rethink fig 1. I don't get the meaning/purpose of figure 1. I think it is related to my point above, i.e., showing the diversity of processes contained in the remote-sensing based turnover observations but it did not help me. Turning this figure into a table may help. After reading the entire manuscript, I think I would have benefited more from a description of each of the terms with an example rather than the bars and arrows.

>> We think that the reviewer is referring to Table 1? This table is aimed at showing how different processes can contribute to changes in turnover and biomass both in models and the real world. It is not aimed especially at remote sensing-based estimates. We have made several modifications to the table following the comments of Reviewer 2 in order to make the examples more intuitive and better explain the meaning of the bars. We have also added the following explanation to the introduction:

*"The individual mechanisms within these groupings are isolated within Table 1 so as to show how a particular perturbation in NPP, allocation, or turnover rate of woody or soft tissues (e.g. leaves, fine roots and fruits) would affect biomass or $\tau$. Because both trees and ecosystems respond to environmental stimuli in a coordinated fashion, it is likely that many of these mechanisms will occur in concert."*

(3) Thinning the results section. In my opinion the model comparison is the least developed part of the manuscript and I even doubt whether it is essential. If the definition gets better developed, it might be possible to derive the hypothesis from the definition and then discuss these hypothesis in the light of scientific literature. This would change the type of study but it could increase the impact of this study. If you decide to keep the model comparison, please, better justify the model experiment.

>> Our study is predicated on understanding why TBMs differ in their estimates of turnover time. Although Friend et al. (2014) identified divergence in turnover time projections for the 21st century, we are not aware of a global synthesis of baseline carbon turnover times nor a thorough assessment of the reasons for their divergence (as described in the second paragraph of the introduction). As such

we consider the model intercomparison to be a fundamental component of the study. It is consistent with the Model-Data Synthesis approach advocated by Medlyn et al. (2015), that TBMs can be used to identify hypotheses for further testing by observations. Although all of the hypotheses could have been formulated without the model intercomparison we would have had no indication of their importance for driving differences in turnover time estimates. We have revised paragraph 2 of the introduction to include the following sentences:

*"Relatively little attention has focused on the representation of τ and its drivers in current vegetation models, with some but not all relevant dependencies represented in different models."*

*"The divergence that can be traced to TBM structure and parameterisation (Nishina et al., 2015) has not been closely analysed in terms of the contributions of specific underlying processes, interactions and driver dependencies, or their basis in knowledge from real world ecosystems."*

(and add revision numbers for each of the models).

>> We will add revision or version numbers for each of the models to Table 2.

It would have been much easier to compare the models if a run with a prescribed PFT distribution was used as well.

>> Prescribed PFT distributions are only possible for a subset of models here (listed as average-individual in Table 2). Cohort- and individual-based models compete PFTs directly against each other within the same stand both in terms of vertical competition for light and horizontal competition for water and, in some models, nutrients. As such, their PFT distributions are emergent outcomes of the model simulation and cannot be fixed without fundamentally altering simulated stand structure.

How can you justify the comparison of data with management to simulations without management? How meaningful is this given that management is a major driver of both the growth and the mortality components of turnover?

>> Our study concentrates on comparing between TBMs (please see below), which made simulations using a consistent protocol. We have made comparison to satellite-based turnover times as an independent comparison, recognising that they also include management (line 243).

Including management in the simulations would have broadened the scope of the study from the topic of natural vegetation dynamics and greatly complicated the interpretation of an already complex response. We agree that management is an important driver of forest biomass turnover, however, to keep the task to a manageable level of complexity we explicitly exclude management from the scope of the study (please see lines 131-133).

Given the complexity of the processes described by turnover but the simplicity of the observations (i.e. a single number), the model comparison remains superficial in the sense that it is hardly possible to label some of the model behavior as "very unlikely". In the end this section takes up a lot of space for very little information (although I liked Fig 2 a lot. It is an informative way to show both models and data – note that this is the only figure that shows the observations).

>> Figures 1 and 2 both show the satellite-based method for estimating turnover time, although we would hesitate to call these data observations, as satellite-based NPP products derive NPP using a modelling approach and large-scale biomass products rely on empirical modelling approaches to extrapolate in space. We have included observations or alternative approaches where possible, but note that large-scale observational constraints on biomass turnover time are extremely limited. It is this lack of observations which has led to such a diversity of approaches and outcomes in TBMs. As

new products become available in the future it should indeed become possible to identify unlikely model behaviour. In the absence of these products, we have focused on decomposing the reasons behind the model responses (encapsulated in the hypotheses) and outlining how uncertainty might be reduced in the future.

Maybe the bulk of the comparison could be moved to the supplementary materials?

>> Much of the comparison is already shown in the supplementary materials. We prefer to keep most of the existing figures in the main text as they each underline important results relating directly to the derivation of the hypotheses. We have modified the selection of figures in the main text slightly, following suggestions of Reviewer 1.

**References**

Friend, A. D., Lucht, W., Rademacher, T. T., Keribin, R., Betts, R., Cadule, P., Ciais, P., Clark, D. B., Dankers, R., Falloon, P. D., Ito, A., Kahana, R., Kleidon, A., Lomas, M. R., Nishina, K., Ostberg, S., Pavlick, R., Peylin, P., Schaphoff, S., Vuichard, N., Warszawski, L., Wiltshire, A. and Woodward, F. I.: Carbon residence time dominates uncertainty in terrestrial vegetation responses to future climate and atmospheric $CO_2$., Proc. Natl. Acad. Sci. U. S. A., 111(9), 3280–3285, doi:10.1073/pnas.1222477110, 2014.

Medlyn, B. E., Zaehle, S., De Kauwe, M. G., Walker, A. P., Dietze, M. C., Hanson, P. J., Hickler, T., Jain, A. K., Luo, Y., Parton, W., Prentice, I. C., Thornton, P. E., Wang, S., Wang, Y., Weng, E., Iversen, C. M., Mccarthy, H. R., Warren, J. M., Oren, R. and Norby, R. J.: Using ecosystem experiments to improve vegetation models, Nat. Publ. Gr., 5(6), 528–534, doi:10.1038/nclimate2621, 2015.

Nishina, K., Ito, A., Falloon, P., Friend, A. D., Beerling, D. J., Ciais, P., Clark, D. B., Kahana, R., Kato, E., Lucht, W., Lomas, M., Pavlick, R., Schaphoff, S., Warszawaski, L. and Yokohata, T.: Decomposing uncertainties in the future terrestrial carbon budget associated with emission scenarios, climate projections, and ecosystem simulations using the ISI-MIP results, Earth Syst. Dyn., 6(2), 435–445, doi:10.5194/esd-6-435-2015, 2015.